# ON THE INTERVENTIONAL CONSISTENCY OF AUTOENCODERS

## ABSTRACT

Autoencoders have played a crucial role in the field of representation learning since its inception, proving to be a flexible learning scheme able to accommodate various notions of optimality of the representation. The now established idea of disentanglement and the recently popular perspective of causality in representation learning identify modularity and robustness to be essential characteristics of the optimal representation. In this work, we show that the current conceptual tools available to assess the quality of the representation against these criteria (e.g. latent traversals or disentanglement metrics) are inadequate. In this regard, we introduce the notion of *interventional consistency* of a representation and argue that it is a desirable property of any disentangled representation. We develop a general training scheme for autoencoders that takes into account interventional consistency in the optimality condition. We present empirical evidence toward the validity of the approach on three different autoencoders, namely standard autoencoders (AE), variational autoencoders (VAE) and structural autoencoders (SAE). Another key finding in this work is that differentiating between information and structure in the latent space of autoencoders can increase the modularity and interpretability of the resulting representation.

## 1 INTRODUCTION

Representation learning is the problem of finding a low dimensional description of the data. The characteristics of a 'good representation' have long since been a matter of debate, often depending on the context in which the representation has to be employed. Multiple works (cf. Bengio et al. (2013), Schölkopf et al. (2021)) identify *modularity*, *robustness* to distribution shifts and *interpretability* as crucial features of a 'good representation', motivating the quest toward disentanglement (Bachman et al. (2019), Locatello et al. (2020), Ridgeway & Mozer (2018)) and causal representations (Suter et al. (2019), Träuble et al. (2021)). The insight shared by both these fields is that complex world phenomena arise from the 'rich interaction of many sources', and thus enable compact descriptions in terms of the basic components participating in their generative process. More formally, the hypothesis is that there exist a set of semantically meaningful variables $S$ and an arbitrarily non-linear function $G$ such that $\mathbf{X} = G(S_1, ..., S_n)$, where $\mathbf{X}$ are the observations.

Autoencoders have played a crucial role in the field of representation learning since its inception. Their success is largely due to their simplicity and surprising effectiveness. Mathematically autoencoders can be represented as the tuple $(E : \mathbb{R}^d \rightarrow \mathbb{R}^n, D : \mathbb{R}^n \rightarrow \mathbb{R}^d)$. Capacity constraints (e.g. $n \ll d$) force the latent space to prioritize certain information in the input, thus yielding a useful representation. Additional requirements may be imposed either structurally or through regularisation. Intuitively, consistency means that an input generated by the decoder can be mapped back to the point in the latent space that produced it. The central idea of Cemgil et al. (2020) is making the encoder and the decoder consistent both on the training data $P(X)$ and on the auxiliary observations generated by the decoder $P(X')$.

**Our contribution.** Unsupervised causal representation learning is largely unsolved today. In a recent paper Locatello et al. (2019) prove that 'the unsupervised learning of disentangled representations is fundamentally impossible without inductive biases on both the models and the data'. Motivated by this result, in this work we, investigate the structure and in particular the self-consistency

of the learned autoencoder, rather than focusing on the representation's relation to the "groundtruth" factors. In this context, we formulate an inductive bias for autoencoders, that we call *interventional consistency*, and we link it to the solution space of the disentanglement problem. Additionally, we introduce a new architectural module for the latent space of autoencoders which unifies the disentanglement objective and the, more structured, causal perspective.

## 2 METHOD

Consider the *'artificial'* generative process implemented by the decoder as $X \sim P(Z)P(X|Z; \theta)$. We express the latent space dynamics in terms of a Structural Causal Model (SCM) (Pearl, 2009) on $Z$. An SCM is defined by a set of so-called *noise* variables $N = N_1, ..., N_n$, with a distribution $P(N)$ that factorises, and a set of *structural assignments* $f_1, ..., f_n$ of the form:

$$Z_i := f_i(\text{PA}_i, N_i) \quad \text{for } i = 1, ..., n, \tag{1}$$

where $\text{PA}_i$ refers to the set of *direct causes* of the variable $Z_i$. The set of directed interactions between causal variables identifies a graph, which is usually assumed to be acyclic (i.e. a DAG). This formulation naturally entails a distribution $P(Z)$ and a corresponding *causal factorisation*:

$$P(Z) = \prod_i P(Z_i|PA_i). \tag{2}$$

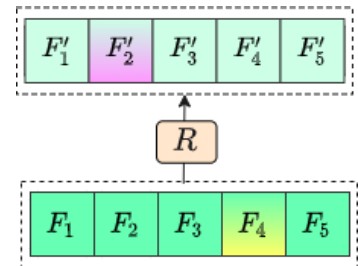

Figure 1: Diagram illustrating *Interventional consistency*: a change in one of the factors participating in $P(Z)$ (i.e. $F_4$) affects only one of the factors in $P(Z')$ (i.e. $F'_2$). The response $R$ maps $Z$ to $Z'$.

For a thorough introduction to SCM we refer the reader to Peters et al. (2017). We can now apply results from causality to the artificial generative process. In particular, we turn to the ICM principle (Peters et al., 2017), which states that the modules participating in the generative process are independent and can thus be separately manipulated. Let us refer to the composition $R = E \circ D$ as the *response map*, as done in Leeb et al. (2021), mapping the latent space $\mathcal{Z} \subseteq \mathbb{R}^n$ back to itself, i.e. $R : \mathcal{Z} \rightarrow \mathcal{Z}$. Let $R(Z'|Z)$ be the posterior distribution entailed by the map, with $Z, Z'$ being distinct random variables. Furthermore, consider the *response aggregate posterior* distribution and any factorisation thereof:

$$P(Z') = \int P(Z)R(Z'|Z)dZ = \prod_i P(Z'_i|Z'_{j \neq i}).$$

The implications of the ICM principle are trivial if applied to the prior $P(Z)$: modifying $P(Z_i|PA_i)$ will not affect $P(Z_j|PA_j)$ for $j \neq i$. However the consequences get far from trivial if we require the response map to preserve the validity of the principle. Let $\tilde{P}(Z)$ denote the *intervention distribution*:

$$\tilde{P}(Z) = \tilde{P}(Z_m|\tilde{PA}_m) \prod_{i \neq m} P(Z_i|PA_i). \tag{3}$$

Then we say that the ICM principle is preserved by the response map if:

$$\tilde{P}(Z') = \int \tilde{P}(Z)R(Z'|Z)dZ = \tilde{P}(Z'_k|Z_{j \neq k}) \prod_{i \neq k} P(Z'_i|Z'_{j \neq i}), \tag{4}$$

where the indices $m$ and $k$ are not necessarily equivalent, although we will from here on assume their equivalence.[1] Intuitively, $R$ preserves the ICM principle if any localised change in $P(Z)$ produces a corresponding localised change in $P(Z')$. The idea is illustrated in Figure 1. Crucially, this property mirrors the definition of disentanglement in the context of the artificial generative process: each factor in the representation ($P(Z'_k|Z'_{j \neq k})$) is sensitive to changes in a single generative factor ($P(Z_m|PA_m)$), while being relatively invariant to changes elsewhere in the generative process. This

---

[1]Allowing $m \neq k$ essentially takes into account the possibility of permutation of the original space dimensions in the representation, which is the norm in disentanglement metrics. By assuming $m = k$ we get rid of one degree of freedom in the problem.

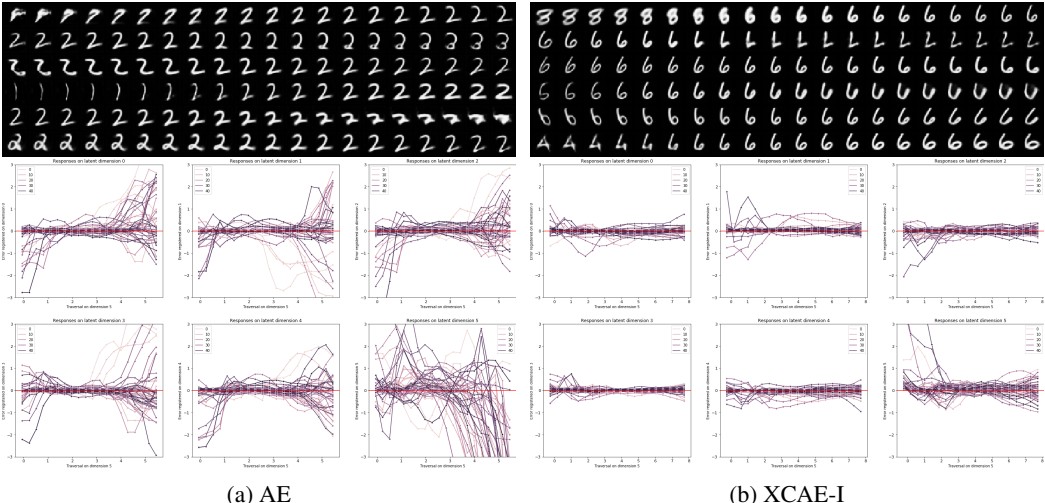

(a) AE                                       (b) XCAE-I

Figure 2: (Top) Traversals for autoencoder models on MNIST. (Bottom) Response error measured on each of the six latent dimensions when traversing along the last dimension ($= 5$). Each line (in different colors) represents a sample drawn from the aggregate posterior. The horizontal axis tracks the dimension being traversed. The vertical axis records the difference between response and original sample along the corresponding dimension. For unregularized models, the traversals are performed in a range whose extremities are identified by the aggregate marginal posterior $Q(Z)$. Observe that the model equipped with consistency training exhibits significantly lower error upon intervention than a standard autoencoder.

analogy suggests that the condition expressed in Equation 4 has to be satisfied by a representation that disentangles the true generative process. We call this condition *interventional consistency*.

Interventional consistency can be formulated in terms of invariance and equivariance of the response map. If the autoencoder satisfies interventional consistency, then $P(Z'_i|Z'_{j<i})$ is invariant to the action of the group of interventions $\mathcal{I}_m$ localised on $P(Z_m|Z_{j<m})$ with $i \neq m$. Moreover, assuming we are able to perform an intervention on the response variables $I'$ equivalent to the one performed on the prior $I$, the response map is equivariant to the intervention, meaning that the order in which the response and the intervention are applied is not important. From this symmetry viewpoint, interventional consistency belongs to causal representations: 'the right causal order is the one invariant to the right kind of interventions' (Herb Simon, cf. Hoover (2008)).

Importantly, a representation that satisfies the interventional consistency condition does not necessarily solve the identifiability problem (Gresele et al., 2021), as any bijective transformation of the true causal factors would equally satisfy the condition for the correspondingly transformed interventions. Instead, it is *manipulable* by construction. Without this feature, there is no guarantee the decoder can extrapolate to latent samples not seen during training. As an example, consider the common diagnostic tool of latent traversals: it consists of atomic interventions applied to the different dimensions of the representation. Often a good representation is associated with semantically meaningful traversals in the image space. However, the observed result is only half of the story: if the encoder fails to interpret the generated samples the representation cannot be called disentangled. To illustrate this example, in Figure 2 we show the error recorded on the response to traversals on a standard autoencoder.

## 2.1 CONSISTENCY TRAINING

It is possible to use the invariance or equivariance formulation to incorporate interventional consistency into the solution space of an autoencoding problem. Let $\mathcal{L}_{\text{INV}}(R)$ and $\mathcal{L}_{\text{EQV}}(R)$ be real valued functions measuring the violation of the invariance and equivariance property by the response map $R$, respectively. Then the space of interventionally consistent autoencoders is defined as:

$$\{(E, D) : \mathcal{L}_{\text{INV}}(E \circ D) = 0 \text{ or } \mathcal{L}_{\text{EQV}}(E \circ D) = 0\}.$$

This hard constraint is replaced with a differentiable regularization by augmenting the optimization objective with $\mathcal{L}_{\text{INV}}$ or $\mathcal{L}_{\text{EQV}}$. Next, we propose evaluation metrics that make the consistency penalties easier to interpret. Algorithm 1 and 2 in the Appendix demonstrate how to compute $\mathcal{L}_{\text{INV}}$ and $\mathcal{L}_{\text{EQV}}$. Importantly, $\mathcal{L}_{\text{INV}}$ is independent of the network architecture of choice, therefore it can be applied to any existing autoencoder method with latent space $N$. Meanwhile, $\mathcal{L}_{\text{EQV}}$ only requires an explicit distinction between noises $N$ and causes $Z$ (of the artificial generative process) outlined in the previous section, which, in our experiments, is implemented by the explicit causal latent block (see 4).

Note some additional mild approximations are necessary to make the penalty terms tractable. Firstly, we assume that the response aggregate posterior $P(Z')$ factorises like $P(Z)$, i.e. that the statistical dependencies in the prior are preserved by the response map. When using an autoencoder with explicit structural mappings this assumption is equivalent to assuming that the response aggregate *noise* posterior $P(N')$ factorises. Secondly, we employ Monte Carlo sampling to estimate the consistency by sampling hard interventions on the noise variable distribution of the form: $\tilde{P}(N_m) \leftarrow \delta(v)$, with $v$ from the aggregate marginal posterior $Q(N_m) = \mathbb{E}_X Q(N_m|X)$. Finally, in order to assess equivariance we treat $I' = I$, which relies on the equivalence between $Z$ and $Z'$. This assumption is satisfied as long as the reconstructions are sufficiently similar to corresponding the training samples (i.e. the autoencoder fidelity is high).

Below, $N^k$ denotes a sample from the prior $P(N)$ and $\tilde{N}^k$ its intervened version. Moreover we let $\bar{N}'^k$ and $\bar{\tilde{N}}^{k\prime}$ be the expectation of the response to $N^k$ and $\tilde{N}^k$, respectively. Let $\hat{\bar{N}}^{k\prime}$ be the result of the intervention $I$ applied to $\bar{N}'^k$ and we denote the corresponding causal variables by $\hat{\bar{Z}}'^k$ and $\bar{\tilde{Z}}^k$ .

**Invariance score.** Let $U_{I_m}$ be a measure of the magnitude of the intervention $I_m$ and $\sigma_i$ the standard deviation of each dimension in the response space. Then we define the invariance error in response to the intervention $I_m$ recorded on the dimension $i$ as

$$e_{I_m,i} = \frac{1}{U_{I_m}\sigma_i}\mathbb{E}_N\big[||(\bar{N}' - \bar{\tilde{N}}'_{I_m})_i||_2\big].$$

More precisely we define $U_{I_m} := \mathbb{E}_N\big[||(N - \tilde{N}_{I_m})_i||_2\big]$. Consequently we score the invariance of the $i$-th dimension with respect to interventions on the $m$-th dimension as:

$$\mathbf{INV}[m,i] = 1 - \mathbb{E}_{I_m \sim \mathcal{I}_m}\big[e_{I_m,i}\big]. \tag{5}$$

We approximate all the expectations with their Monte Carlo estimates. According to this formulation, a perfectly invariant response map would score 1 on the off-diagonal elements of the matrix. Notice that invariance is not concerned with the diagonal entries of the matrix. However, we can interpret a high invariance score on the $i$-th diagonal element as a sign of the unresponsiveness of the dimension as it implies interventions on $i$ are not registered in the response. We henceforth refer to these unresponsive latent dimensions with the appellative "collapsed", in reference to the well-known effect of posterior collapse (rather common in $\beta$VAE models).

**Equivariance score.** Similarly, we define the equivariance error and equivariance score as :

$$r_{I_m,i} = \frac{1}{U}\mathbb{E}_N\big[||(\hat{\bar{Z}}'_{I_m} - \bar{\tilde{Z}}'_{I_m})_i||_2\big]$$
$$\mathbf{EQV}[m,i] = 1 - \mathbb{E}_{I_m \sim \mathcal{I}_m}\big[r_{I_m,i}\big].$$

In the case of equivariance perfect consistency corresponds to all the entries being 1. Notice however that high equivariance is also obtained by the trivial solution (i.e. a constant encoding map). This is due to the fact that the intervention values are sampled from the aggregate posterior.

**Self-consistency score.** Finally, the self-consistency score evaluates the response inner consistency under no intervention. More specifically, we measure the amount of error introduced by the response on average over the prior. In formula:

$$\mathbf{SCN}[i] = 1 - \frac{1}{\sigma_i}\mathbb{E}_N\big[||(\bar{N}' - N)_i||_2\big]$$

Importantly, self-consistency is the only score relating the prior and the response space and is closely related to the regularization used by AVAEs (Cemgil et al., 2020).

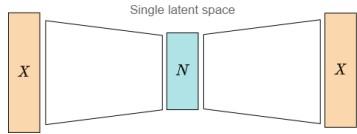 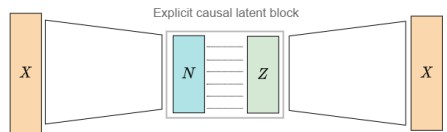

Figure 4: Diagram comparing the structure of an autoencoder with (right) and without (left) the Explicit Causal Latent Block. Instead of a single space modelling the representation, an XBlock employs a noise or information space $N$ and a causal or structure space $Z$.

## 2.2 EXPLICIT CAUSAL LATENT BLOCK

Inspired by the distinction between noise and structure in the SCM we insert a similar bias into the latent space of autoencoders by directly joining the representation layer with a structural mixing block as shown in Figure 4. We henceforth will refer to the botteneck layer as noise terms or $N$, and to the units produced by the subsequent mixing layer as causes or $Z$. Each $Z_i$ is obtained from its corresponding noise term $N_i$ and its predecessors $Z_{<i}$ through a learned nonlinear function $f_i$, much like in an SCM (implemented as an MLP). To increase the sparsity of the structure we apply a learnable mask $M_i \in \mathbb{R}^{(i-1)}$ to the parents , such that $Z_i = f_i(M_i \odot Z_{<i-1}, N_i)$. The masks are parametrised through the Gumbel trick (Jang et al., 2016) to induce weight sharpening, and thus act as a gating mechanisms of the information provided by the predecessors.

We name this extension of the representation layer *"explicit causal latent block"* (*XBlock*) and we call *XNet* (or prefix "X") any network using this module. Most disentanglement methods attempt to learn a representation consisting of causal variables or disentangled "factors" which are statistically independent (Higgins et al., 2017a). However, except in the trivial case, it is not the $Z_i$ that should be treated as statistically independent, but the $N_i$. Consequently, the explicit separation between noise and structure reconciles with the conventional disentanglement perspective while also providing the model with additional power to learn identifiable relationships between the latent variables terms. The causal links revealed by $M_i$ can then be used to help identify how interventions will affect the resulting samples for downstream tasks such as controllable generation. Moreover, differentiating between noise and structure allows to expand the dimensionality of the causal variables $Z$, without any need to reparametrise the noise distribution. We partition the causes in units, each one consisting of multiple dimensions and representing a single statistical variable.

## 3 RELATED WORK

To the best of our knowledge, this is the first work to formally introduce the notion of *interventional consistency* and develop a training scheme based on it. However, related ideas have been discussed in previous works.

Recently, there is particular interest in applying causality to representation learning (Kocaoglu et al. (2017), Suter et al. (2019), Träuble et al. (2021), Schölkopf et al. (2021) ). In Yang et al. (2021) the authors introduce a learnable *Causal Layer*, which essentially describes a SCM, in the latent space of a variational autoencoder. Much like our XBlock, the causal layer in Yang et al. (2021) transforms independent noise variables into causal ones, thus separating the informa-

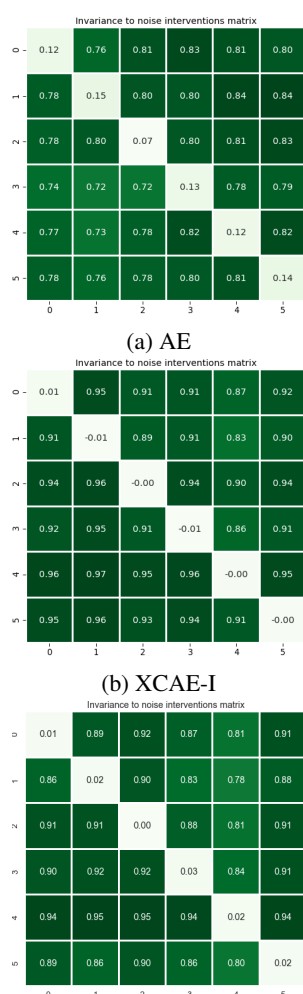

Figure 3: Invariance matrices for autoencoders of the Standard (a), 'XC-I' (b) and 'XC-E' (c) variant. We observe a pronounced increase in the score due to the consistency training. Note that invariance and equivariance regularisation achieve a similar performance.

tion from the structure. The structural assignments of the Causal Layer integrate physical limitations to the system's dynamics that are not present in the XBlock (i.e. additive noise assumption, mild non-linearity). Moreover, Yang et al. adopt a perspective radically different from this work: their goal is to solve causal disentanglement, and they rely on weak supervision to do so.

The introduction of symmetries in the representation layer has been examined from multiple perspectives (Cohen & Welling (2016), Finzi et al. (2020), Finzi et al. (2021), Weiler et al. (2018), Benton et al. (2020), **?**). In Keller & Welling (2021) the authors use topographic capsules parametrised through variational inference to obtain equivariance to geometric transformations in the input. Like Yang et al. (2021), the work uses weak supervision (in the form of ordered input sequences) to disentangle salient characteristics in the data. Interestingly, in Keller & Welling (2021) disentanglement is obtained as a by-product of equivariance, echoing the definition of disentanglement given in Higgins et al. (2018).

Perhaps the discussion most relevant to the present work is presented by Leeb et al. (2021) and Besserve et al. (2018). Leeb et al. develop a framework to assess the characteristics of the response map of variational autoencoders, thus laying the groundwork of the present study. The invariance score proposed in Section 2.1 is closely related to the *Latent Response Matrix* presented in Leeb et al. (2021). Unlike this paper, in Leeb et al. additionally introduce new tools to evaluate representations in regards to disentanglement based on the response map. In Besserve et al. (2018), interventions on the deep layers of a generative model are employed to discover independent modules participating in the representation. Similarly to the present work, Besserve et al. (2018) adopts a perspective completely agnostic of the true generative process instead focusing on the artificial one. However both Leeb et al. and Besserve et al. are only concerned with analyzing and evaluating generative processes, while we develop training methods and metrics to directly optimize for the desired structure.

## 4 EXPERIMENTS

In the following experiments we demonstrate the relevance of interventional consistency and the strength of the proposed approach. More specifically, we are able to learn representations that are markedly more robust to interventions and can correctly handle manipulations. We quantitatively and qualitatively verify that consistency training is effective in improving the model's interventional consistency without harming the model's generative performance. Moreover we produce evidence on the benefit of differentiating between information and structure in the latent space, by analysing the effects of the XBlock on the learned representation.

### 4.1 EXPERIMENTAL DETAILS

In our experiments we employ three baselines with fundamental differences in their representation layer, namely: convolutional autoencoders (*AE*) (Bourlard & Kamp (1988), Hinton & Zemel (1994)), variational autoencoders ($\beta VAE$) (Higgins et al., 2017b), and structural autoencoders (*SAE*) (Leeb et al., 2020). For each of the baselines two variants are considered: *'X'* and *'XC'*, where the former extends the original model by integrating the explicit causal latent block and the latter augments the 'X' version through *consistency training*. The 'XC' variant in turn comes in two modalities employing the invariance ('XCI') and equivariance ('XCE') regularisation respectively. Additionally, we include an implementation of the AVAE model to compare the two consistency training schemes.

The experiments are performed on MNIST (LeCun & Cortes, 2010) and CelebA (Liu et al., 2015) datasets. These datasets have been chosen for being closer to real-world scenarios than artificially manufactured alternatives whilst being well known and historically recognised in the literature.

The neural architectures considered vary only in size, keeping a coherent overall skeleton across experiments. Specifically, we consider 3 different size standards for MNIST (*'S', 'Standard', 'S4'*) and another 2 for CelebA (*'v32', 'v324'*). The differences are reported in Table 3 in the appendix, together with the details on the skeleton and training hyperparamters.

## 5 RESULTS AND DISCUSSION

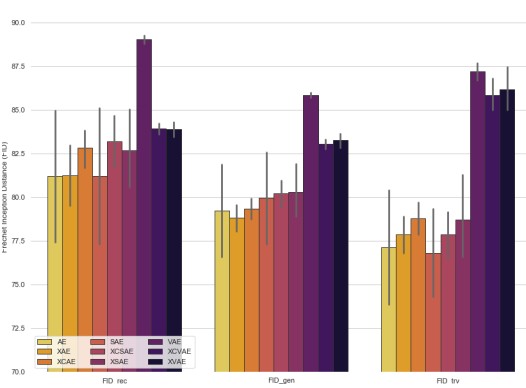

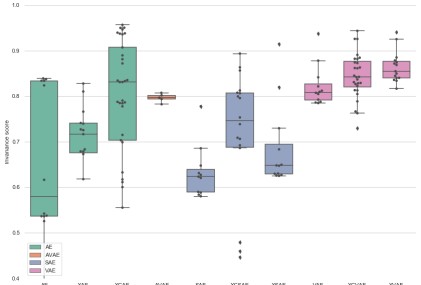

(a) FID scores for all the models on MNIST (lower is better). We evaluate the FID on the reconstructions (left), prior samples (middle) and traversals (right). Perhaps unsurprisingly, the variational baseline scores markedly higher than the alternative. Crucially, the 'X' and 'XC' versions' scores are comparable to the Standard models.

(b) Summary of the invariance score on all experiments (MNIST and CelebA). For each model, all its variants are evaluated against the invariance score. The matrix is reduced to a single number through averaging. The variational baseline scores systematically higher than the alternatives. This result is partly due to the presence of collapsed dimensions in the representation, which increase the model's score due to their absolute unresponsiveness. Importantly, we notice that the 'XC' variants tends to perform better than the others.

**Consistency training improves the model's interventional consistency without harming the reconstruction quality.** From Figure 7 and 6 it can be seen across all datasets that the 'X' and 'XC' variants do not interfere with the quality of the reconstructed or prior samples. To quantitatively support this claim we include the corresponding FID scores (Heusel et al., 2017) for all the experiments in Figure 5a. Additional samples are shown in Section A.4.1. One principle against which to evaluate the quality of the prior samples is their intrinsic diversity. Real world datasets such as MNIST and CelebA are rich in low probability details that are thus hard to capture in the representation. In this regard the autoencoder outperforms the respective VAE models (see A.4.1). The prior samples for deterministic autoencoders are obtained through *hybrid sampling*, a technique proposed in (Leeb et al. (2020), Besserve et al. (2018)) which allows to generate previously unseen combinations of latent factors by randomly aggregating different posterior samples. For more details on hybrid sampling we refer the reader to Leeb et al..

Figure 2 shows the standard latent traversal plot alongside the traversals response error. The error is measured as the signed difference between the intervened prior noise vector and its expected response as defined in Section 2.1, i.e. $\tilde{N}_i^k - \tilde{N}_i'^k$. This figure best depicts the discrepancy between generation and inference which lies at the heart of this work and motivates the consistency perspective. By only analysing the traversal plots one might conclude that the learned representation can correctly resolve the alteration introduced by the traversal, and thus that it possesses the generalisation and modularity suggested by the quality of the generated samples. In short, consistency is assumed to hold. However, the errors recorded on the response show that this assumption is false: the intervention applied is not preserved by the encoder, which presumably projects the sample into the observational distribution manifold (Radhakrishnan et al., 2018). In summary Figure 2 reveals that (i) the current autoencoders (i.e. without consistency requirements) are indeed not consistent under interventions and that (ii) consistency training can improve the model's consistency. Significantly, consistency training only acts on the response error indirectly since the regularisation terms do not include take into account the prior samples. The increase in consistency with consistency training can also be assessed in Figure 3. It must be taken into account that the invariance regularisation directly aims at optimising the invariance score, whilst the equivariance score only does so indirectly. Full summaries of the consistency scores for the different training modalities can be found in Section A.4.3. A quantitative summary of the invariance scores for all experiments is given in Figure 5b. We observe that the AVAE baseline has the lowest score on interventional consistency among the variational family, comparably to the $\beta$VAE. On the contrary it is among the highest

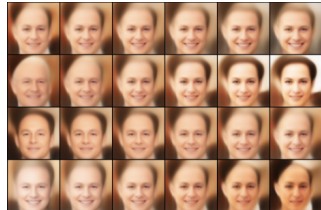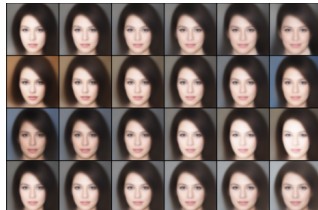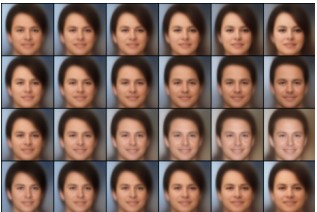

Figure 6: Traversals for the autoencoder model in 3 variants, namely: Standard (left), 'X' (center) and 'XC-I' (right). The different variants are comparable in terms of generation quality. However, the Xnets show a higher degree of modularity between the dimensions. We hypothesise that the Xblock favours independence. We include the full traversal plots in the Additional material (18).

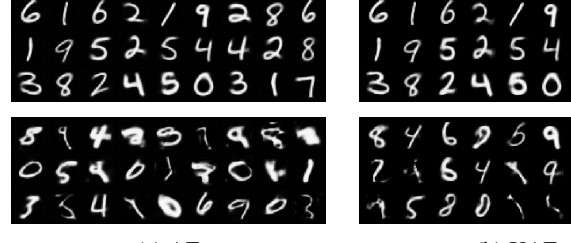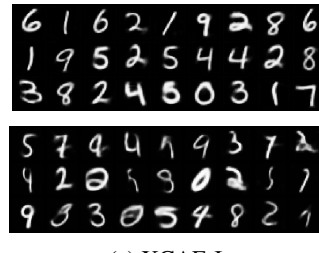

|        (a) AE        |        (b) XAE        |        (c) XCAE-I        |

Figure 7: Input reconstructions (first row) and prior samples (second row) for the autoencoder model in 3 variants, namely: Standard, 'X' and 'XC-I'. The prior samples are obtained through hybrid sampling from the aggregate posterior distribution.

performing models when evaluated on self-consistencty (see Figure 28). This discrepancy confirms that consistency, as defined in Cemgil et al. (2020) does not imply interventional consistency.

**The explicit causal latent block provides sparsity and self-consistency.** A mask element with a low value prevents the information from passing from the parent to the child node. Consequently, if the noise variables are independent, one can read off the adjacency matrix the structure of the existing statistical relations between latent units. This is not possible in the current standard of a single space carrying both the information and the statistical structure. A sparse configuration is desirable for interpretability, since it simplifies the dynamics of the generation process. Moreover simpler models are more likely to be correct, according to the widely accepted *principle of parsimony* also known as *Occam's razor*. Section A.4.4 in the appendix displays the mask values for the learned XBlock of different models.

Table 1 reports the sparsity levels for different latent sizes. The results show that it is possible to obtain sparsity for every model without the need of any additional regularisation. The introduction of learnable masks endowed with weight sharpening has *proved to be sufficient to induce sparsity*

| LATENT SIZE ($N$) | SPARSITY (%) |
| --- | --- |
| 6 | 41.66 |
| 12 | 45.83 |
| 32 | 48.44 |

Table 1: Sparsity levels per varying latent size. The right column records the percentage of edges in the Xblock over the set of possible connections based on the learned mask.

| SAMPLING MODE | AE | XAE | XCAE |
| --- | --- | --- | --- |
| Posterior | **0.96** | **0.96** | 0.95 |
| Hybrid | 0.75 | 0.81 | **0.88** |
| Uniform | 0.66 | 0.68 | **0.80** |

Table 2: Average self consistency under posterior, hybrid and uniform sampling for AE (left), XAE (middle) and XCAE (right). The average is taken over the latent dimension for the S MNIST models.

*in the latent block structure*. The number of latent dimension has systematically turn out to be the only indicator for the sparsity level achieved. Moreover, the structure of dependencies in the model varied only slightly with respect to the model family and exhibited greater sensitivity to the latent unit size (cf A.4.4).

Table 2 records the self consistency of autoencoders under three different prior sampling procedures: posterior, hybrid and *uniform* sampling, respectively from top to bottom. 'Posterior' refers to drawing samples from the aggregate posterior $Q(N) = \int Q(N|X)P(X = x)dx$, 'hybrid' applies to its hybridised version, and finally in the 'uniform' case we consider a uniform distribution whose extremes on each dimension are derived from $Q(N)$. We would normally expect a decreasing self-consistency going from the first to the last row: in fact, hybrid sampling breaks any existing statistical correlation between the noise variables, and uniform sampling additionally compels a continuous support in the latent data manifold to be preserved by the response map. In Table 2 the expected trend is approximately confirmed by the AE model. We notice that the 'X' version improves the self consistency score under all sampling procedures, proving to be less sensitive to the distribution shifts introduced by hybridisation and continuity. Consistency training further increases the score under all sampling modes. This behaviour suggests the hypothesis that the explicit causal latent block in fact favours independence between the noise dimensions.

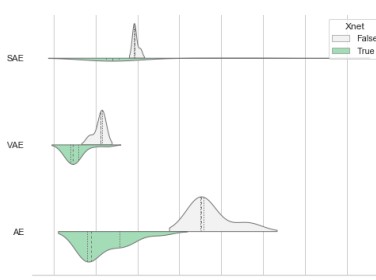

Figure 8: MMD between noises aggregate posterior and hybrid sampling distribution, measured using an Inverse Multiquadratic kernel (IMQ). The distributions are obtained by recording the MMD metric over all S models configurations (including different random seeds).

In order to give an objective ground to this hypothesis in the additional material (Figure 8) we look at an empirical measure of independence between the noise variables for all models. More specifically, we compute the Maximum Mean Discrepancy (MMD) between the aggregate posterior distribution over the noise variables and its hybridised version. By construction hybridisation destroys any existing statistical relationship between the latent dimensions. For both the AE and VAE model families the 'X' version's metric distribution appears significantly shifted to the left, thus indicating a higher degree of independence between the noise variables.

## 6 CONCLUSION

In this work we focus on the modularity and self-consistency of a representation, rather than addressing the identifiability problem. More specifically, we have formulated and studied the interventional consistency property of autoencoders. The metrics and regularization objectives we develop result in representations that remain consistent under interventions thus enabling meaningful manipulation. We propose a novel architecture that can be applied to any existing autoencoder, inspired by causal representation learning, to explicitly model the dependencies between latent variables. The architecture has empirically shown to improve the model's self-consistency and independence in the latent space. The separation of the information and structure of the representation therein connects the disentanglement perspective to the more structured causal framework, thereby also opening the door to new analysis tools to enrich the representation's interpretability and controllability by exposing the learned mechanisms.

## REPRODUCIBILITY STATEMENT

The used datasets are publicly available. Implementation details for all the experiments are included in the Additional material (A.3). A description of the hyperparameters and network architectures used is included in Appendix A.3. As indicated as an option in the author guide `https://iclr.cc/Conferences/2021/AuthorGuide` for sharing the code anonymously, in a first step we will make a comment directly to the reviewers and area chairs with the link to an anonymous repository once the discussion forums open and in the second step make the full repository public once the paper is accepted.

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

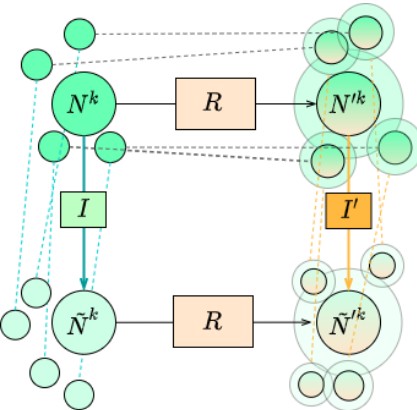

Figure 9: Diagram illustrating interventional consistency.

# A APPENDIX

## A.1 NOTATION

In this section we report a guide to the notation used in the work. Throughout this paper we use $N$ and $Z$ for stochastic variables in the latent space. $N'$ and $Z'$ are used for stochastic variables in the response space. $X$ is be used for input data and $\mathcal{X}$ for the input space. $(E, D)$ denote the encoder and decoder mapping, respectively. $R = E \circ D$ defines the response. $N^k$ denotes a sample from the prior $P(N)$ and $\tilde{N}^k$ its intervened version. Moreover we let $\bar{N}'^k$ and $\bar{\tilde{N}}'^k$ be the expectation of the response to $N^k$ and $\tilde{N}^k$, respectively. For a Dirac posterior (deterministic map) we use its center as expectation. Finally, $\hat{\tilde{N}}^{k\prime}$ marks the result of the intervention $I$ applied to $\bar{N}'^k$ and we denote the corresponding causal variables by $\hat{\tilde{Z}}'^k$ and $\bar{\tilde{Z}}^k$.

## A.2 CONSISTENCY TRAINING

In this Appendix we take a closer look at the connection between interventional consistency and the pointwise estimate of invariance and equivariance. Interventional consistency can be formalised with the two following definitions.

**Definition 1** (*Interventions Invariant Response*). *Given an autoencoder $(E, D)$, and a causal factorisation of the prior over the latent space $P(Z) = \prod_i P(Z_i|PA_i)$ the response $R = E \circ D$ is invariant to interventions on $Z$ if for every intervention $I$ in the set of interventions $\mathcal{I}_m$ localised on the factor indexed by $m$ the following conditions are verified:*

$$P(Z') = \int P(Z)R(Z'|Z)dZ = \prod_i P(Z'_i|Z'_{j\neq i}) \tag{6}$$

$$\tilde{P}(Z') = \int [I \circ P(Z)]R(Z'|Z)dZ = \tilde{P}(Z'_m|\tilde{Z}'_{j\neq m}) \prod_{i\neq m} P(Z'_i|Z'_{j\neq i}) \tag{7}$$

**Definition 2** (*Interventions Equivariant Response*). *Given an autoencoder $(E, D)$, and a causal factorisation of the prior over the latent space $P(Z) = \prod_i P(Z_i|PA_i)$ the response $R = E \circ D$ is equivariant to interventions on $Z$ if, for every intervention $I$ in the set of interventions $\mathcal{I}_m$ localised on the factor indexed by $m$, there exist an equivalent intervention $I'$ acting in the $Z'$ space, such that the following condition is verified:*

$$\int [I \circ P(Z_m|Z_{\tilde{PA}_m})] \prod_{i\neq m} P(Z_i|PA_i)R(Z'|Z)dZ = I' \circ P(Z'_m|Z'_{j\neq m}) \prod_{i\neq m} P(Z'_i|Z'_{j\neq i}) \tag{8}$$

Equivariance implies invariance, as it provides a strictly stronger condition on the response map. The caveat is that to evaluate equivariance one must additionally deduce $I'$ from $I$, which requires

knowledge of the relation between $Z$ and $Z'$. Notice that interventional consistency is effectively a property of the response map and is thus oblivious to the prior $Z$.

Due to the deterministic nature of the structural assignments definitions 1 and 2 can be reformulated without any loss of generality in terms of the noise variables. Let $N$ and $N'$ be the noise variables before and after the response respectively, with

$$P(N') = \int P(N)R(N'|N)dN \tag{9}$$

$$\tilde{P}(N') = \int \tilde{P}(N)R(N'|N)dN = \int \delta(N_m - v) \prod_{i \neq m} P(N_i)R(N'|N)dN \tag{10}$$

In the invariance case we require $P(N')_i$ to match $\tilde{P}(N')_i$ for all $i \neq m$. In the equivariance case we additionally require $\tilde{P}(N')_m = I' \circ P(N')_m = I \circ P(N')_m = \delta(v)$. In formulas, the solution space has to satisfy one of the following conditions:

$$d(\tilde{P}(N')_i, P(N')_i) = 0 \ \forall i \neq m \ \text{ (invariance)} \tag{11}$$

$$d(\tilde{P}(N')_i, I \circ P(N')_i) = 0 \ \forall i \in [1, n] \ \text{ (equivariance)}, \tag{12}$$

where $I \circ P(N')_i = P(N')_i$ for $i \neq m$ and $d$ is a generic distance in the space of distributions on $N$. We can approximate $P(N')$ and $\tilde{P}(N')$ by sampling from the prior (i.e. $P(N)$ and $\tilde{P}(N)$ respectively) and subsequently applying the response map to each sample. Using a pointwise estimate of the above terms can be viewed as adopting a distance that to decompose into the sum of the intra-sample distances. This choice rules out solutions for which the response posterior suffers a shift in a non-intervened dimension when an intervention is applied. We now turn to a full characterisation of the pointwise estimates of invariance and equivariance.

**Pointwise invariance.** We measure the Euclidean distance between the responses expectations, i.e. $||\bar{N}_i'^k - \tilde{\bar{N}}_i^k||_2$ for $i = 1, ..., n$ and $k = 1, ..., T$, with $T$ being the total number of samples. By focusing on the noise terms we can leverage the noise independence assumption to significantly simplify the equations. Notice that we couldn't apply the same direct measurement with the causal variables $\bar{Z}_i'^k$ and $\tilde{\bar{Z}}_i'^k$ because of the existing connections between causal variables, i.e. when modifying $N_m$ we can expect $\bar{N}'_{i \neq m}$ to stay constant but we cannot say the same for $\bar{Z}'_{i \neq m}$ if $i$ is a descendant of $m$. However, invariance can be checked distributionally for $Z$, by looking at the conditionals $P(Z'_i | Z'_{i \neq m})$.

**Pointwise equivariance.** To compute pointwise equivariance we can directly work in the causal variables space measuring the Euclidean distance $||\tilde{\bar{Z}}_i'^k - \bar{Z}_i^k||_2$. The argument for evaluating equivariance at the causal variables level is purely practical: in this way the structural assignments receive a stronger training signal, which is beneficial especially early on in training when the set of connections is being first outlined.

Algorithms 1 and 2 show how to compute pointwise invariance and equivariance, respectively.

**Algorithm 1** Algorithm computing the invariance regularisation term.

---

**for** $m$ *in* $[1, n]$ **do**
  **for** $I_m$ *in* $\bar{I}_m \subseteq \bar{\mathcal{I}}_m$ **do**
    Sample $N^1, ..., N^T \sim P(N)$
    Copy $\tilde{N}^k \leftarrow N^k$ for $k = 1, ..., T$
    Intervene $N_m^k \leftarrow v_{I_m}$
    $e_{I_m} \leftarrow \frac{1}{n \cdot T} \sum_k \sum_{i \neq m} ||\bar{N}_i'^k - \bar{\tilde{N}}_i'^k||_2$
  **end**
**end**
$\mathcal{L}_{INV} = \frac{1}{|\bar{I}_m|} \sum_{I_m \in \bar{I}_m} e_{I_m}$

---

**Algorithm 2** Algorithm computing the equivariance regularisation term.

---

**for** $m$ *in* $[1, n]$ **do**
  **for** $I_m$ *in* $\bar{I}_m \subseteq \bar{\mathcal{I}}_m$ **do**
    Sample $N^1, ..., N^T \sim P(N)$
    Copy $\tilde{N}^k \leftarrow N^k$
    Intervene $N_m^k \leftarrow v_{I_m}$
    Copy $\hat{\tilde{N}}'^k \leftarrow \bar{N}'^k$
    Intervene $\hat{\tilde{N}}_m'^k \leftarrow v_{I_m}$
    $\hat{\bar{Z}}_i'^k \leftarrow f_i(\hat{\tilde{N}}_i^k, PA_i^k)$
    $\bar{\tilde{Z}}'^k_i \leftarrow f_i(\bar{\tilde{N}}'^k_i, PA_i^k)$
    $r_{I_m} \leftarrow \frac{1}{n \cdot T} \sum_k \sum_{i \neq m} ||\hat{\bar{Z}}_i'^k - \bar{\tilde{Z}}'^k_i||$
  **end**
**end**
$\mathcal{L}_{EQV} = \frac{1}{|\bar{I}_m|} \sum_{I_m \in \bar{I}_m} r_{I_m}$

---

The regularisation terms $\mathcal{L}_{INV}$ and $\mathcal{L}_{EQV}$ are included in the standard loss weighted by $\lambda_{INV}$ and $\lambda_{EQV}$, respectively. In our experiments we use $\lambda_{INV} = \lambda_{EQV} = 4$. We consider both stochastic and deterministic encoders, while treating the decoder as deterministic (a Gaussian noise model is used, with the mean parametrised by $D$). If the encoder is stochastic the response $R(N'|N)$ defines a distribution over the $N'$ space: in the case of the VAE this distribution is known to be Gaussian, thus fully described by its mean and variance. Experimentally, including information on the spread of the distribution in the regularisation term has not benefited training, often resulting in numerical instabilities. This outcome has been observed for parametric distances as well as well-known divergence measures (such as the Kullback–Leibler).

## A.3 EXPERIMENTAL DETAILS

In this section we provide the details regarding experimental settings.

All experiments are performed on 1 GPU core NVIDIA A100 40GB and based on the Pytorch (Paszke et al., 2017) framework. The models are trained using the stochastic iterative optimiser Adam (Kingma & Ba, 2014), with default parameters. The learning rate is set to 0.0001 and halved every 30 epochs. The MNIST and CelebA models are trained for 100k and 200k iterations, respectively. We use a batch size of 128 or all the experiments. Each model is trained with 5 different random seeds ($11, 13, 17, 37, 121$) and every pseudo-random number generator in *pytorch*, *numpy* and *python.random* is seeded accordingly.

As anticipated in 4.1 we adopt a single architectural skeleton for all the models, varying only in size. The skeleton is shown in Figure 10. All the architectures use the Mish (Mish, 2020) nonlinearity. The number of pooling layers is set to 3 for MNIST and 7 for CelebA. The number of filters in the convolutional layers is fixed to 32 for MNIST, 100 for CelebA. All the layers, except for the normalization ones, are initialised using the method discussed in Glorot & Bengio (2010). Differently from what is shown in Figure 10 the 'X' and 'XC' variants include an additional layer (forming the explicit causal latent block) applied to the latent vector as shown in Figure 4. Finally, in Table 4 we report the size of each model in terms of the number of trainable parameters.

## A.4 ADDITIONAL RESULTS

### A.4.1 COMPARING SAMPLES QUALITY

In this section we report the reconstructions, generated and traversal samples for the main experiments performed on MNIST and CelebA.

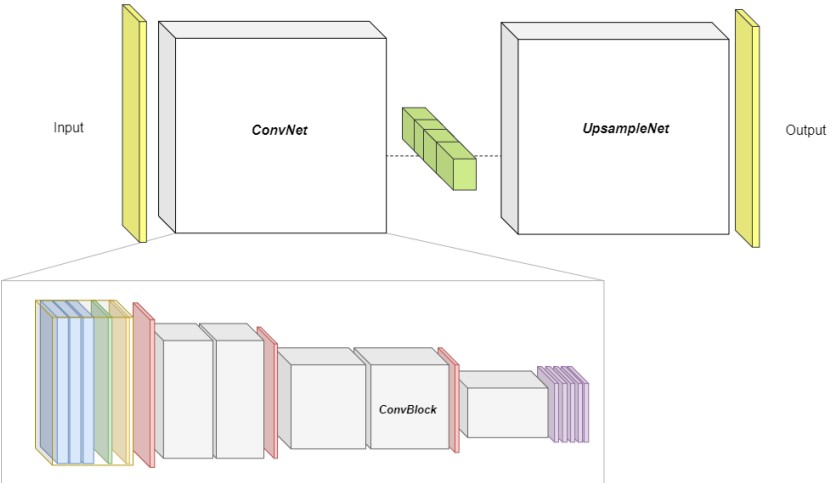

Figure 10: Architectural skeleton used for all models. The input is fed to a convolutional network (*ConvNet*) to obtain its latent representation (in green). This is then fed to a second convolutional network (*UpsampleNet*), equipped with *bilinear upsampling layers*, that produces the output image. The ConvNet comprises blocks of convolutional layers (*ConvBlock*) and regularly spaced *pooling layers* (red). The precise structure of the ConvNet is determined by two parameters specifying the number of ConvBlocks and pooling layers, respectively. As an example, the ConvNet in this figure has depth 6 and 3 pooling layers. Each ConvBlock is composed of a convolutional layer with a fixed number of filters (blue), a non-linear activation (green), and a batch normalisation (yellow). The last ConvBlock in the ConvNet is followed by a *dense head* (purple) which processes the flattened version of the ConvBlock output to reduce it to the latent space size. The dense head consists of 3 linear layers interleaved by non-linear activations. Finally, the UpsampleNet copies the ConvNet structure in reverse order, and with upsampling layers instead of pooling ones. The output of the UpsampleNet is passed through a *Sigmoid* activation to project is back to the input space.

| NAME | DEPTH | LATENT SIZE ($N$) | UNIT SIZE ($Z_i$) | DATASET |
|---|---|---|---|---|
| **Standard** | 12 | 12 | 1 | MNIST |
| **S** | 12 | 6 | 1 | MNIST |
| **S4** | 12 | 6 | 4 | MNIST |
| **v32** | 16 | 32 | 1 | CelebA |
| **v324** | 16 | 32 | 4 | CelebA |

Table 3: Overview of the architecture variants. The Depth entry refers to the number of convolutional layers employed in the encoder. The decoder has depth equal to that of the encoder.

| SIZE | NAME | NUMBER of PARAMETERS |
|---|---|---|
| **Standard** | AE | 423,259 |
| **Standard** | XAE | 425,677 |
| **S** | AE | 309,973 |
| **S** | XAE | 310,984 |
| **S4** | XAE | 313,936 |
| **v32** | AE | 2,840,827 |
| **v32** | XAE | 2,850,299 |
| **v324** | XAE | 2,878,523 |
| **Standard** | VAE | 298,771 |
| **Standard** | XVAE | 301,189 |
| **S** | VAE | 186,025 |
| **S** | XVAE | 187,036 |
| **S4** | XVAE | 188,836 |
| **v32** | VAE | 2,763,707 |
| **v32** | XVAE | 2,773,179 |
| **v324** | XVAE | 2,795,259 |
| **Standard** | SAE | 442,267 |
| **Standard** | XSAE | 444,685 |
| **S** | SAE | 320,488 |
| **S** | XSAE | 320,488 |
| **S4** | XSAE | 323,440 |

Table 4: Number of trainable parameters for all the models in the experiments. The 'XC' model variants have the same size as the 'X' variants and thus are not reported in this table. The training time varies between 3 hours (for the smaller models) to 2 days (for the bigger ones, with consistency training).

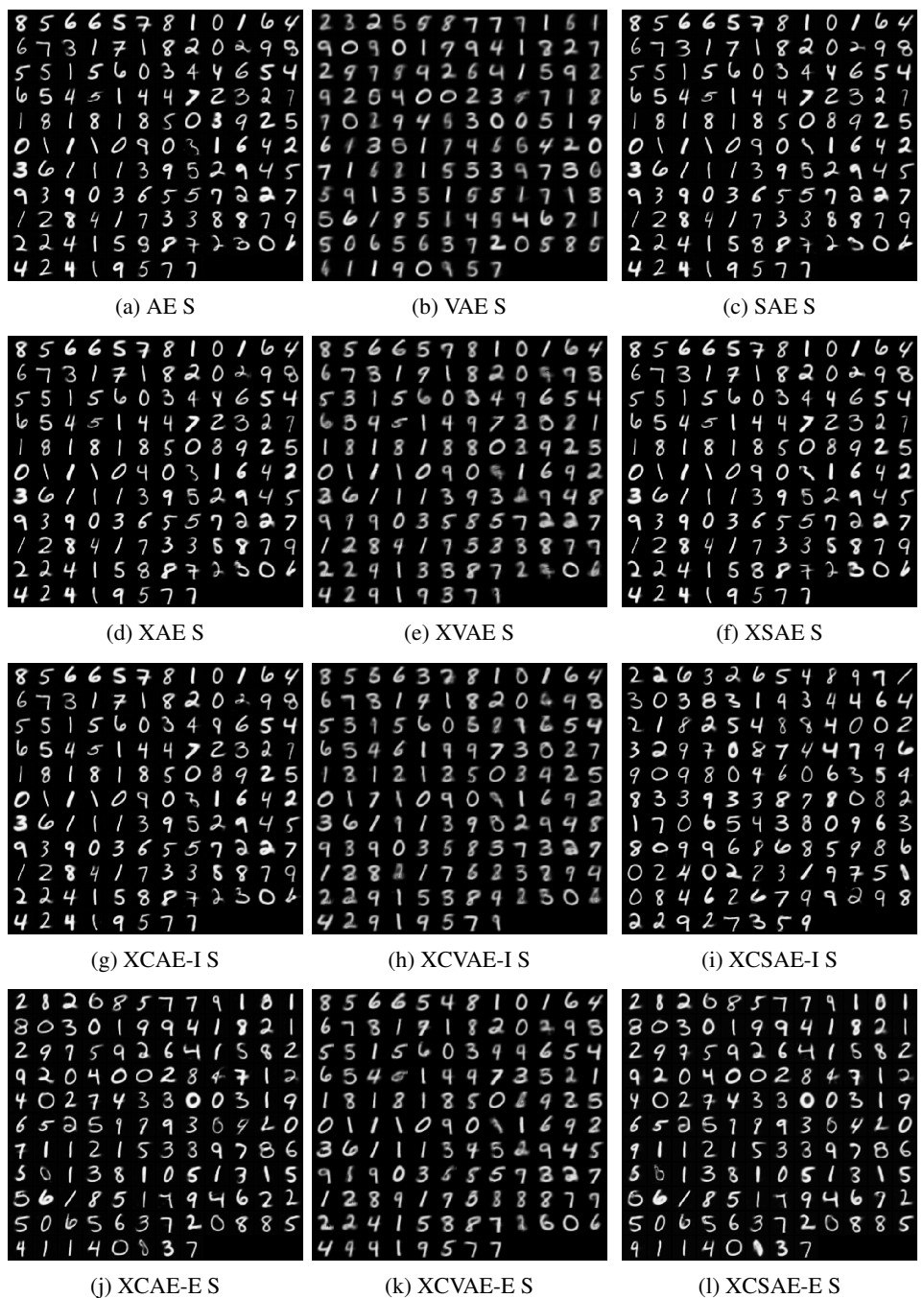

Figure 11: Comparison of reconstruction quality of S models.

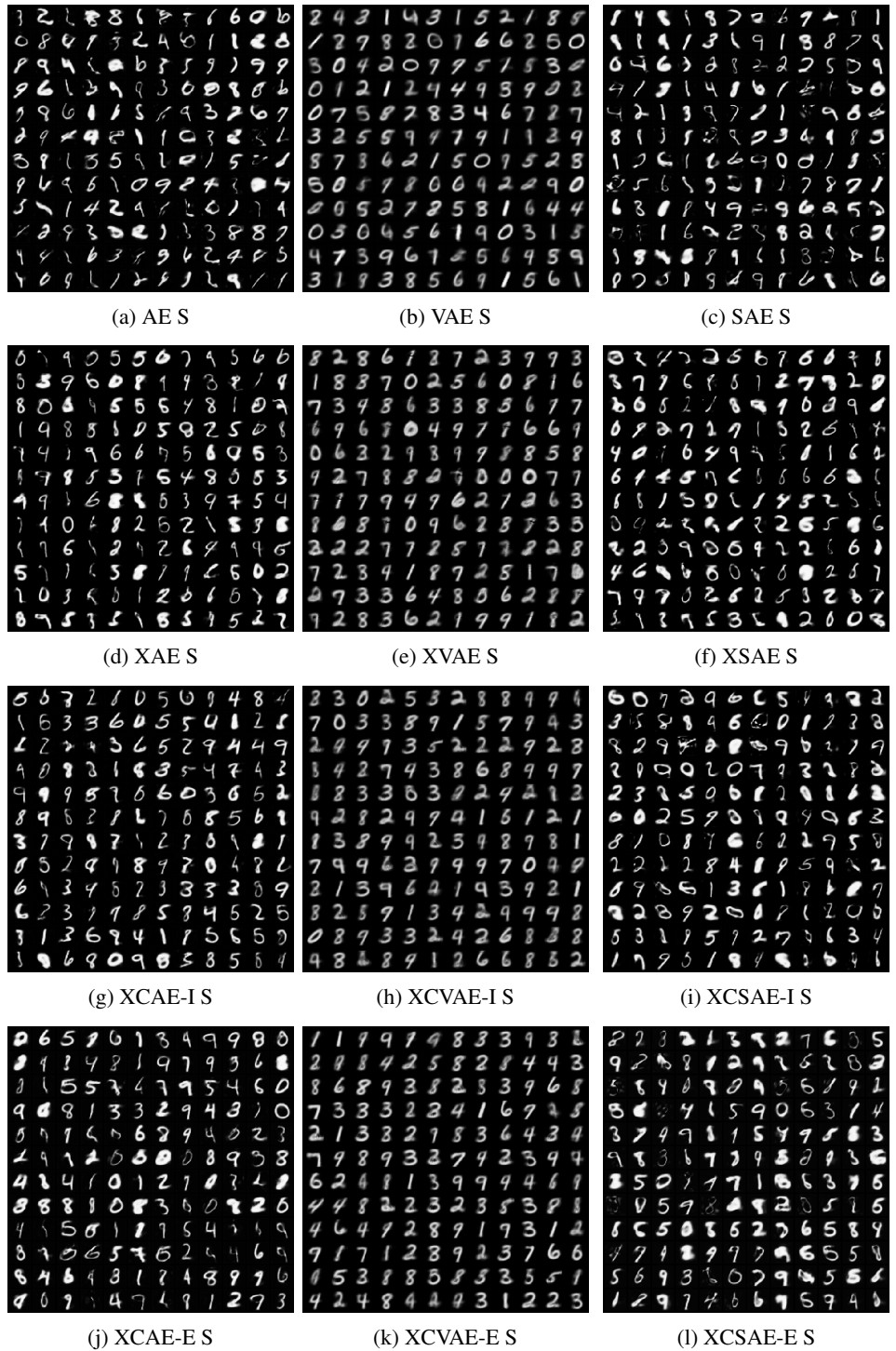

(a) AE S     (b) VAE S     (c) SAE S

(d) XAE S     (e) XVAE S     (f) XSAE S

(g) XCAE-I S     (h) XCVAE-I S     (i) XCSAE-I S

(j) XCAE-E S     (k) XCVAE-E S     (l) XCSAE-E S

Figure 12: Comparison of S models on generated samples quality.

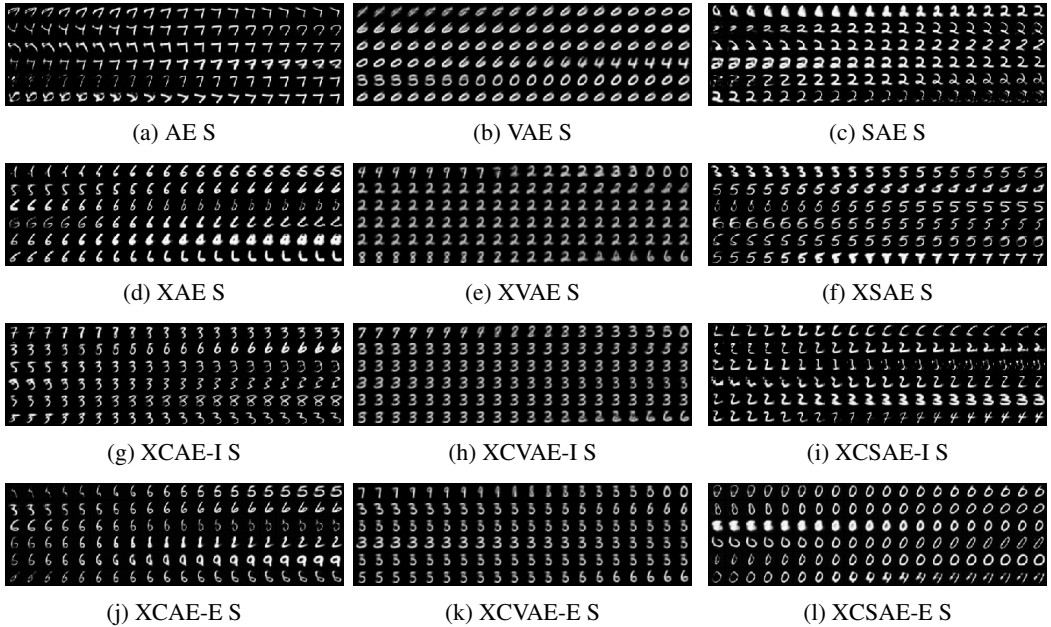

(a) AE S        (b) VAE S        (c) SAE S

(d) XAE S        (e) XVAE S        (f) XSAE S

(g) XCAE-I S        (h) XCVAE-I S        (i) XCSAE-I S

(j) XCAE-E S        (k) XCVAE-E S        (l) XCSAE-E S

Figure 13: Comparison of S models on traversals quality.

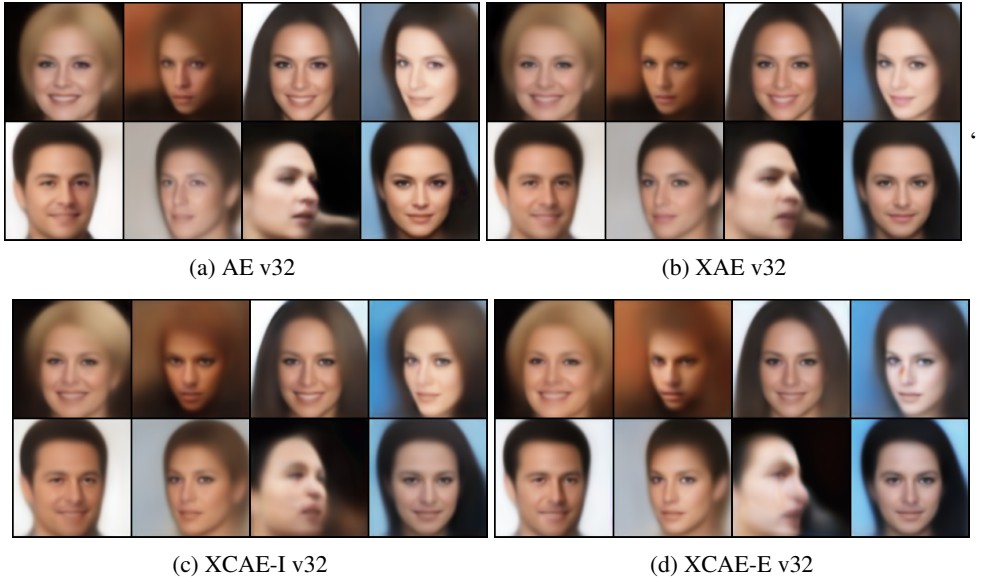

(a) AE v32                  (b) XAE v32

(c) XCAE-I v32              (d) XCAE-E v32

Figure 14: Comparison of v32 models on reconstruction quality.

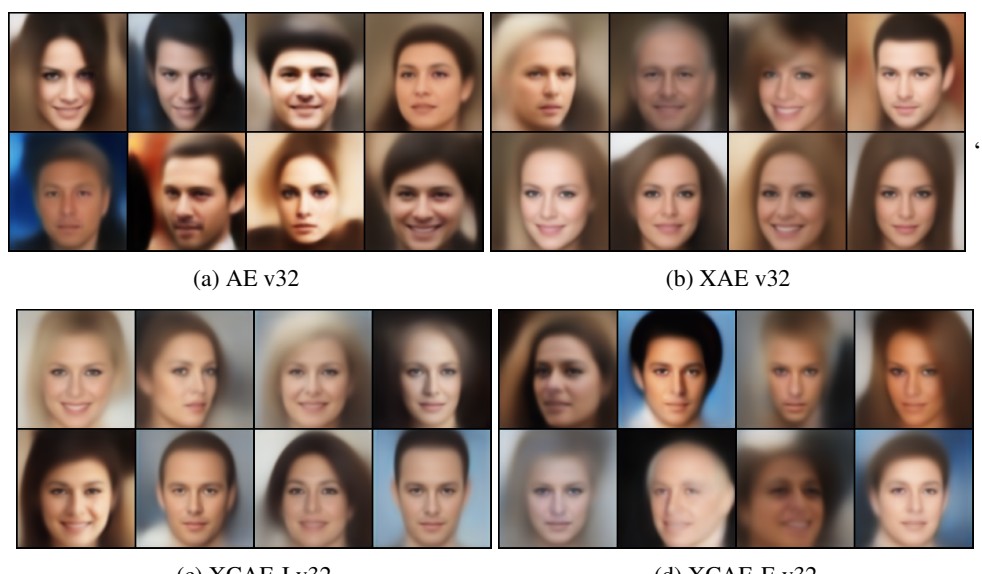

(a) AE v32             (b) XAE v32

(c) XCAE-I v32             (d) XCAE-E v32

Figure 15: Comparison of v32 models on the generated samples quality.

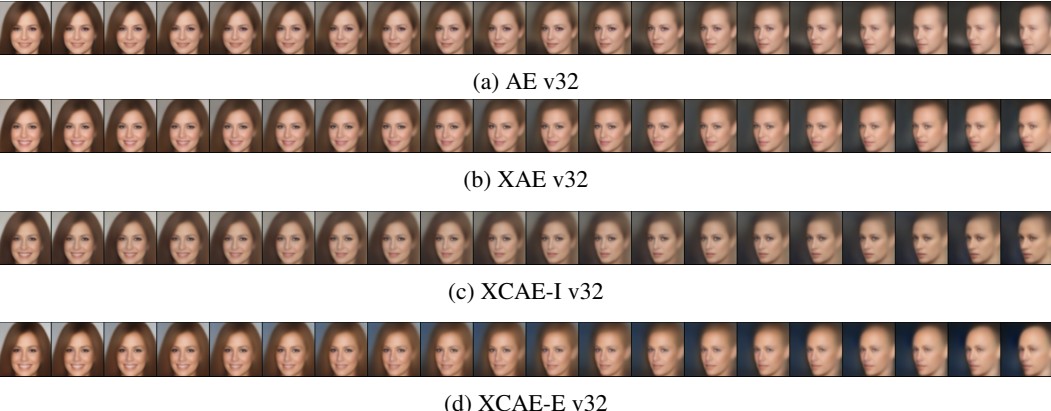

(a) AE v32

(b) XAE v32

(c) XCAE-I v32

(d) XCAE-E v32

Figure 16: Comparison of v32 models on the interpolation samples quality.

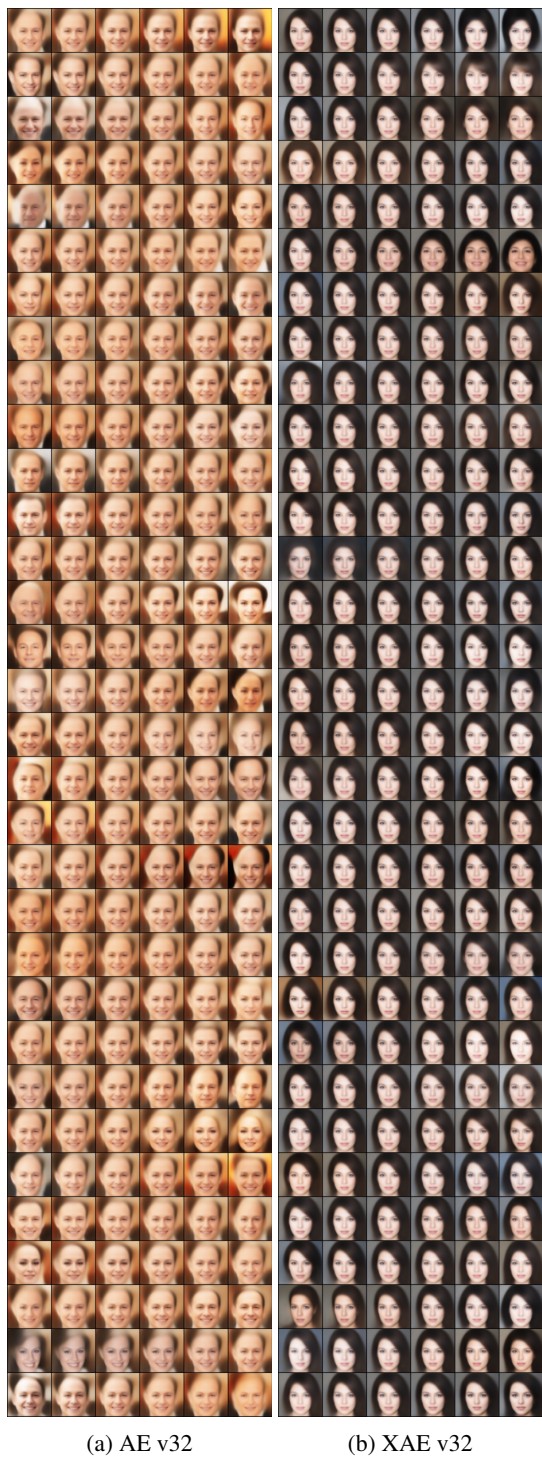

(a) AE v32        (b) XAE v32

Figure 17: Comparison of v32 models on traversals quality (1).

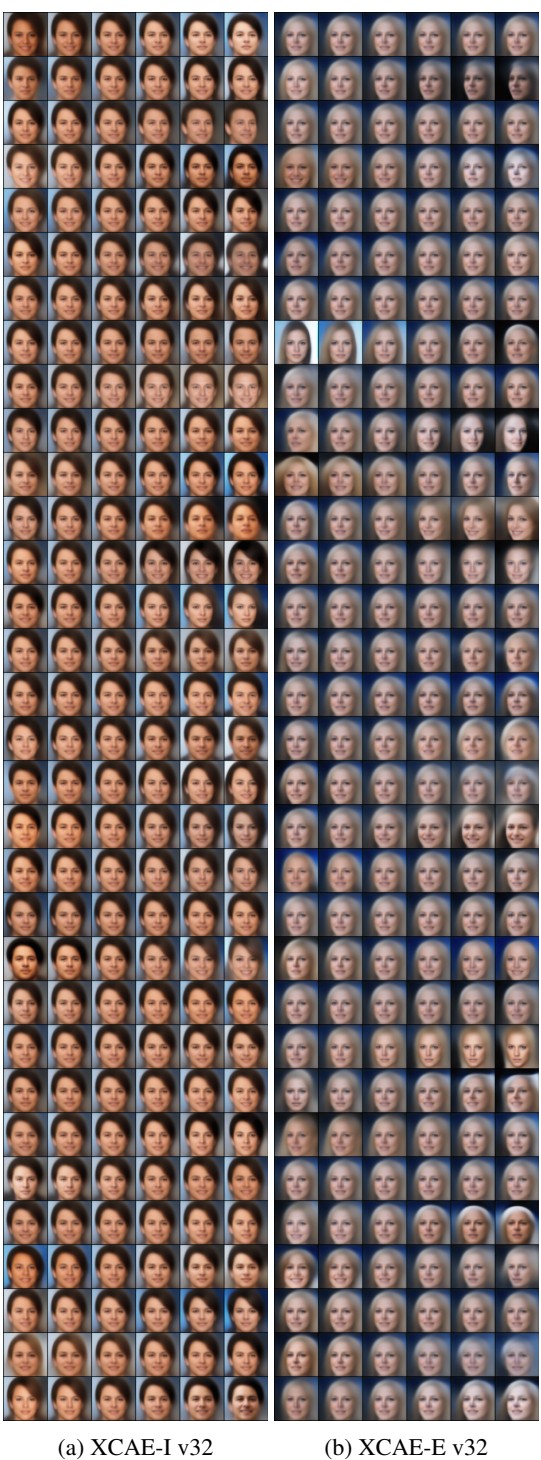

(a) XCAE-I v32          (b) XCAE-E v32

Figure 18: Comparison of v32 models on traversals quality (2).

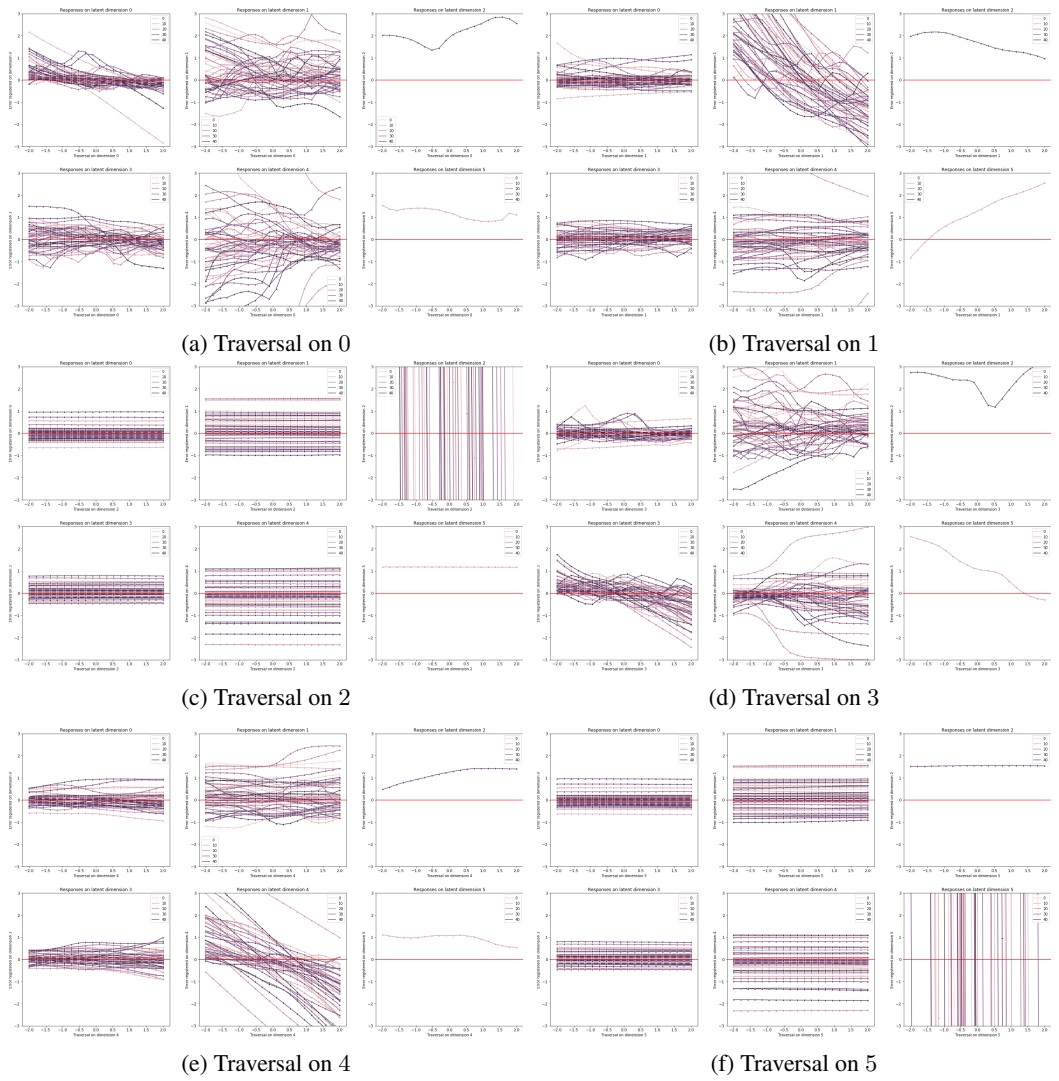

Figure 19: BetaVAE traversal response errors.

### A.4.2 TRAVERSAL RESPONSES.

We here report the complete traversal response plots for the S version of the MNIST experiments. Each plot shows the errors registered on all the latent dimensions as the traversal is performed.

$\beta$**VAE family traversal response errors.** The models in $\beta$VAE family are exposed to the well-known effect of posterior collapse. In the following plots collapsed dimensions can be distinguished by the particular behaviour of the response error (zero when the dimension is not intervened on and equivalent to the intervention value elsewhere). We also include the traversal response errors measured on the AVAE model. Interstingly, all the variants in the VAE family present two collapsed dimensions, except for the XVAE model, that has 3 collapsed dimensions (50% of the total number in this case) and the AVE model which has none.

**AE family traversal response errors.** Additionally we show the traversal response errors for the autoencoder family of models. Notice how in this case, as for the variational case, consistency training effectively decreases the error.

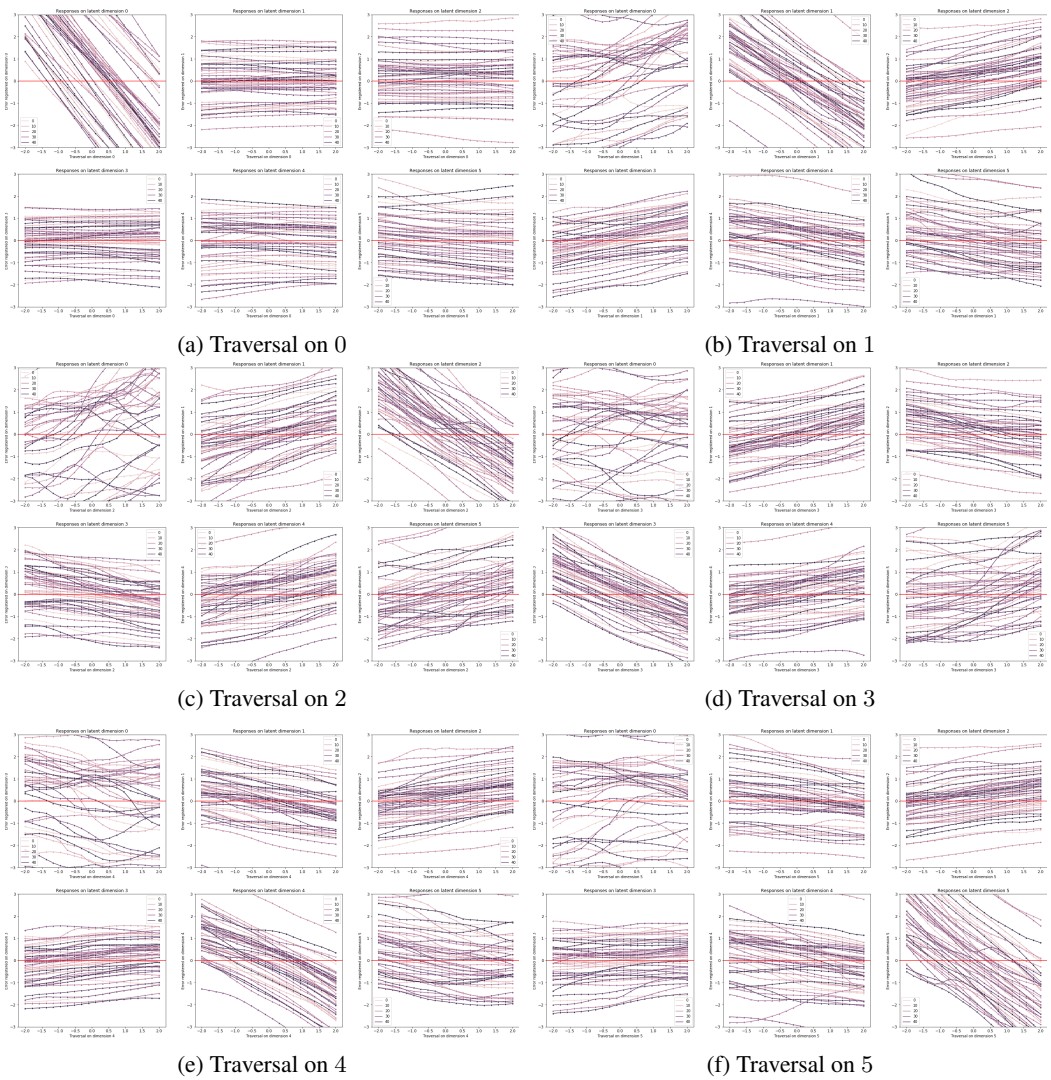

(a) Traversal on 0

(b) Traversal on 1

(c) Traversal on 2

(d) Traversal on 3

(e) Traversal on 4

(f) Traversal on 5

Figure 20: AVAE traversal response errors.

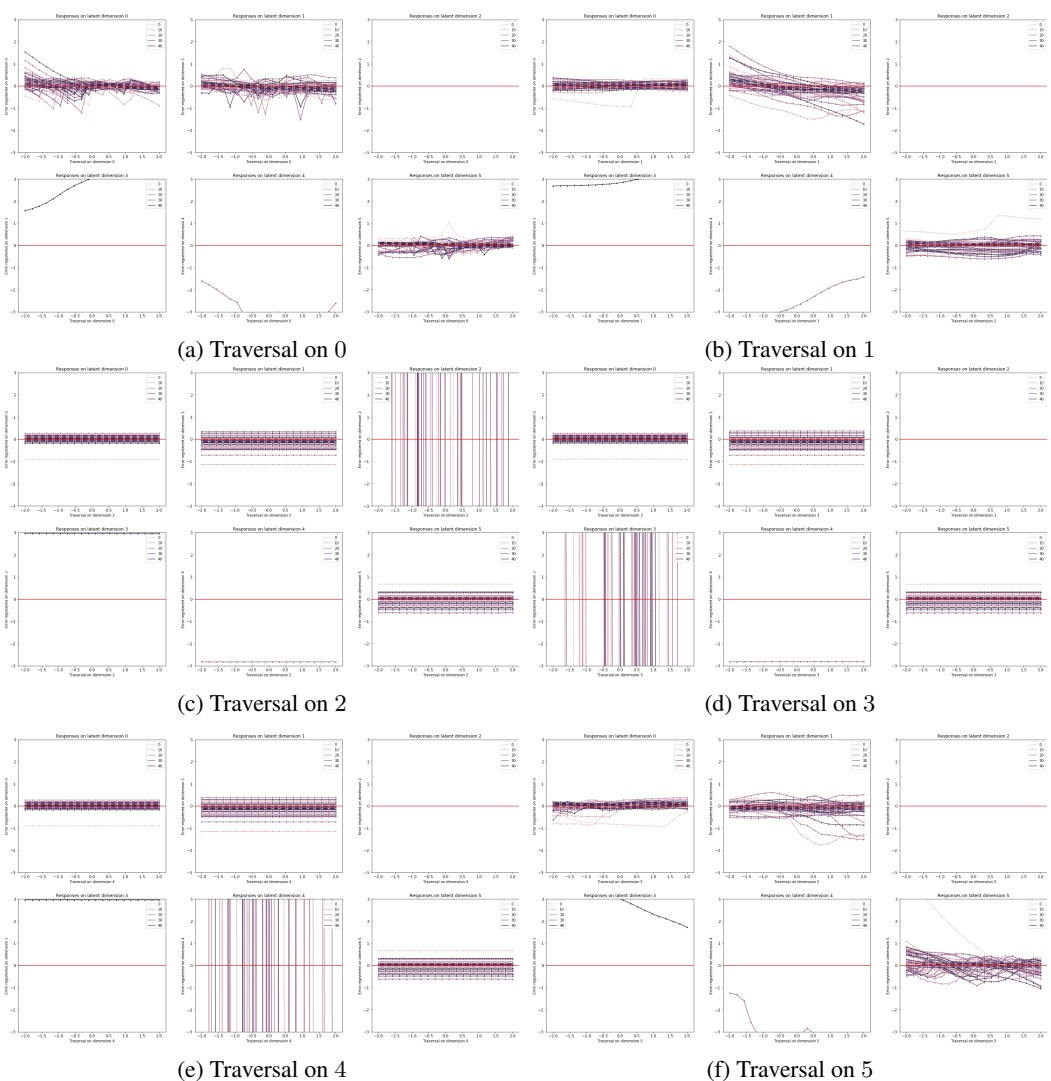

Figure 21: XVAE traversal response errors.

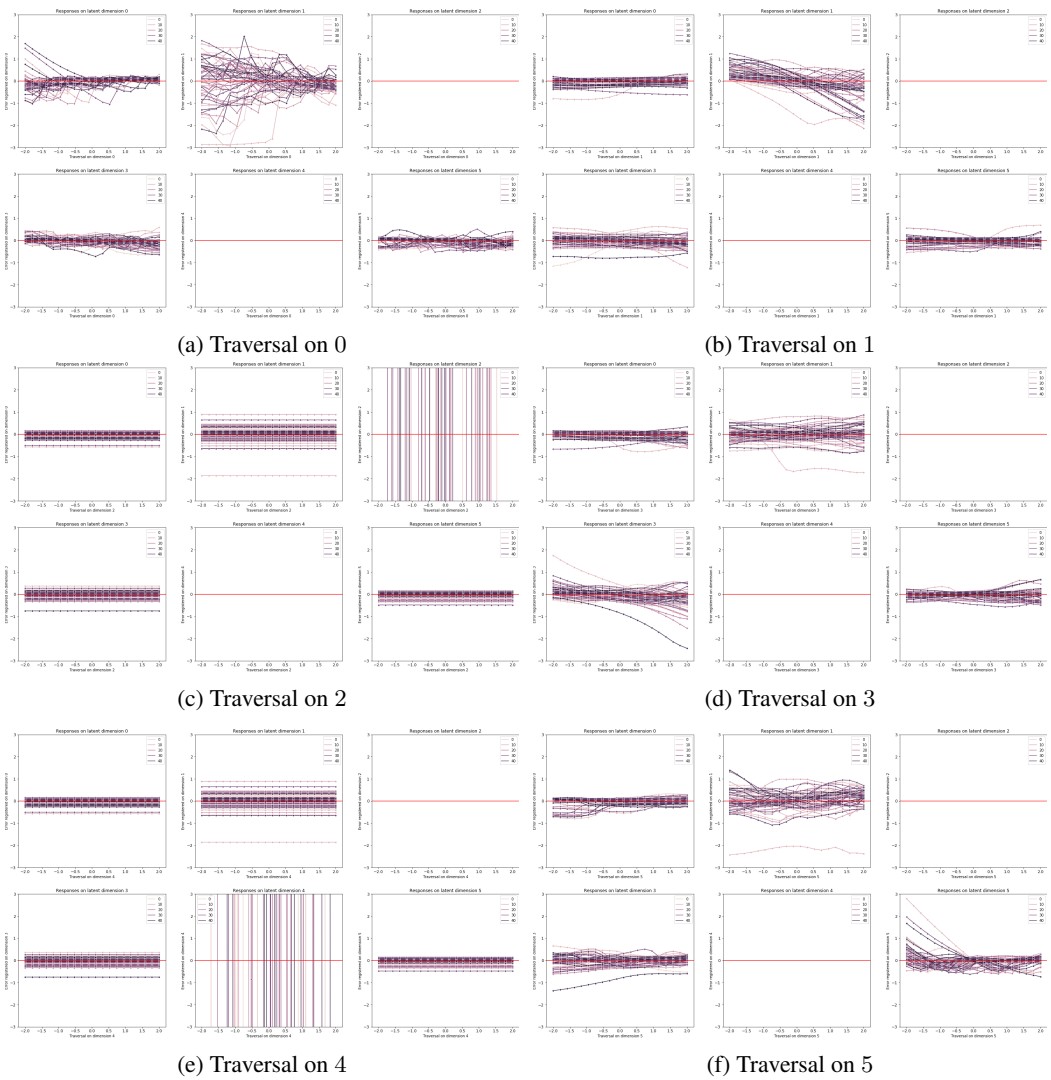

Figure 22: XCVAE-I traversal response errors.

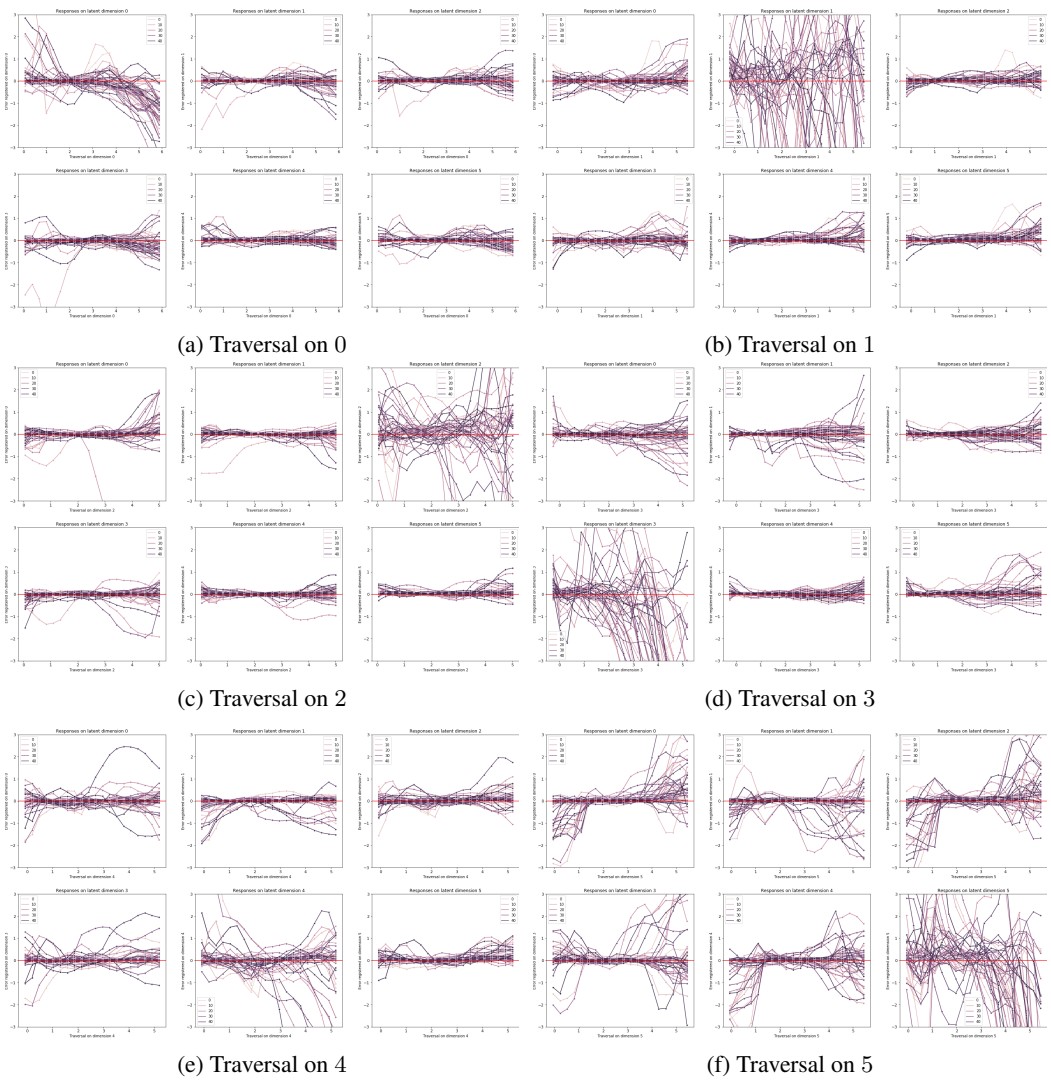

(a) Traversal on 0      (b) Traversal on 1

(c) Traversal on 2      (d) Traversal on 3

(e) Traversal on 4      (f) Traversal on 5

Figure 23: AE traversal response errors.

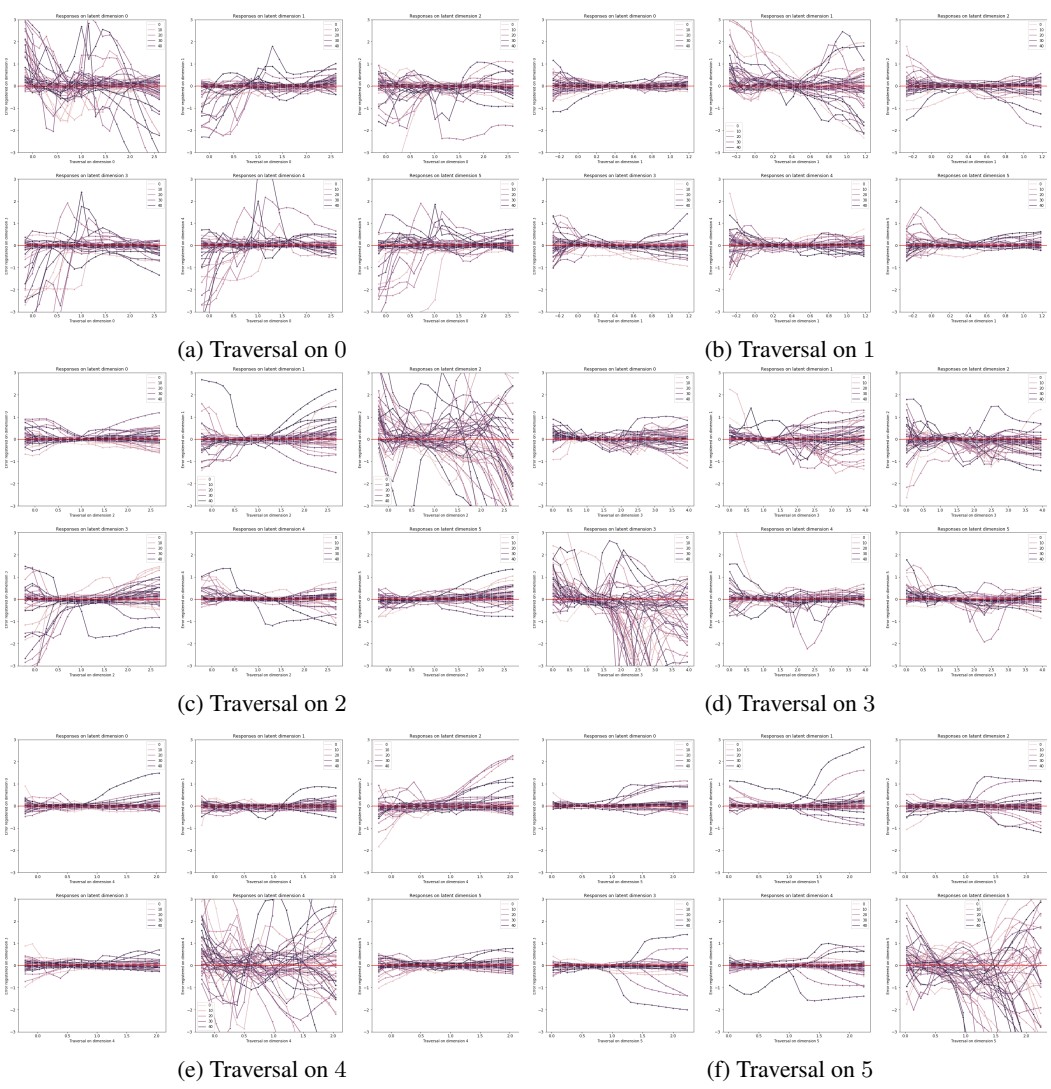

Figure 24: XAE traversal response errors.

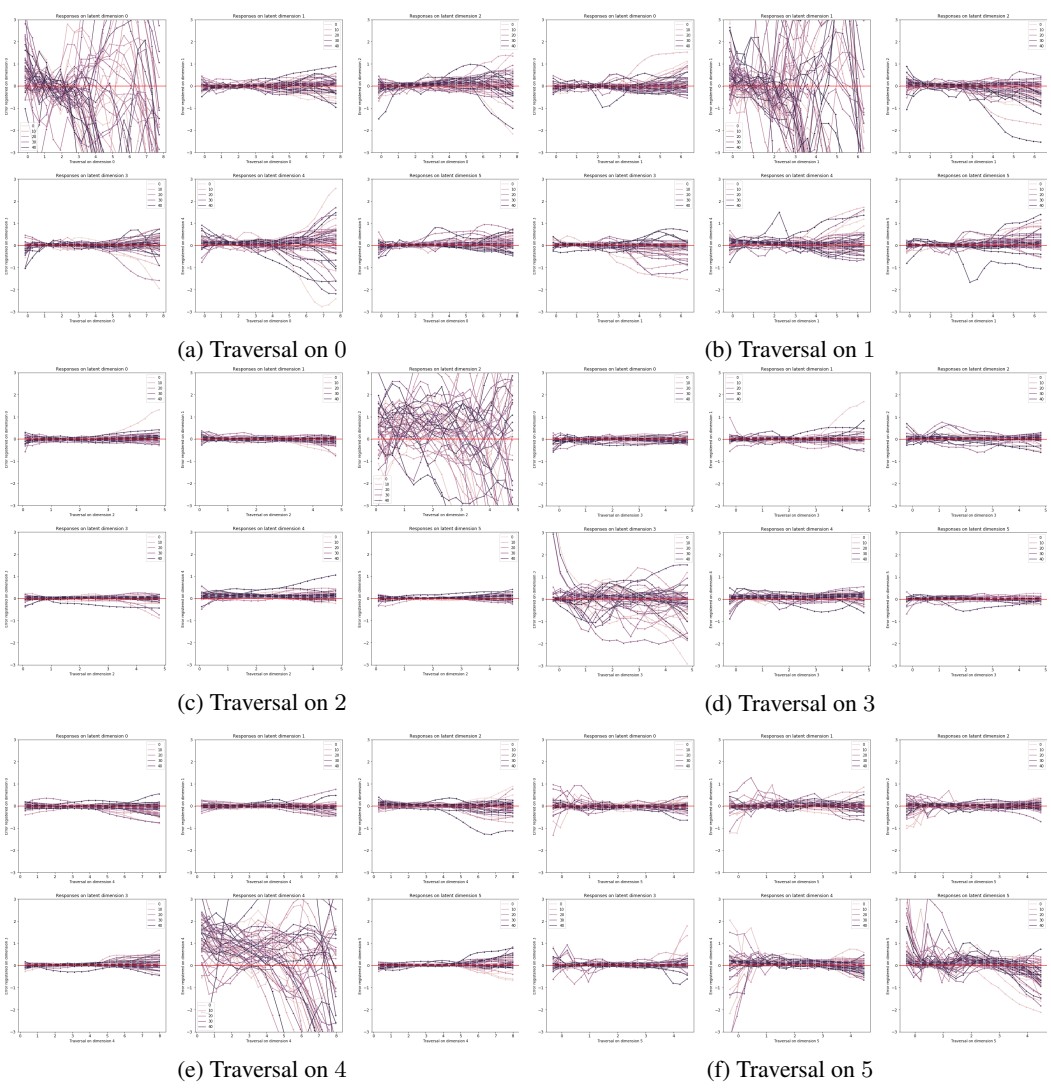

Figure 25: XCAE-I traversal response errors.

### A.4.3 CONSISTENCY METRICS

In this sections we report the results on the new consistency metrics for all the models. We make the following key observations: (i) that consistency training generally strengthens the models' invariance and equivariance, (ii) that although equivariance implies invariance, models using invariance regularisation have comparable performances to those using equivariance regularisation.

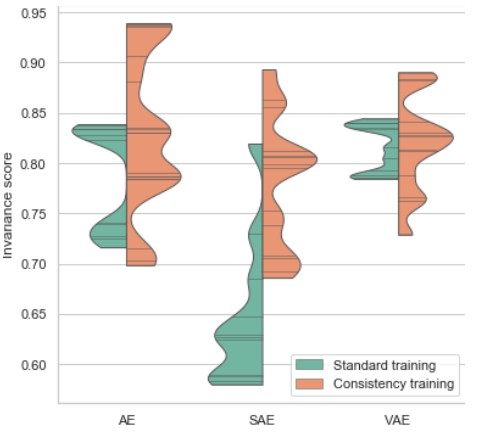

(a) Invariance score for all experiments comparing performance of models with and without consistency training.

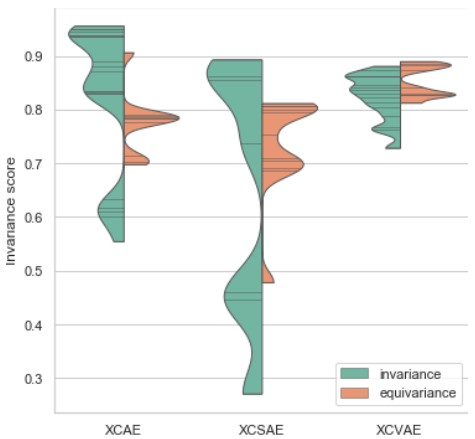

(b) Invariance score for all the models with consistency training comparing invariance-based and equivariance-based regularisation.

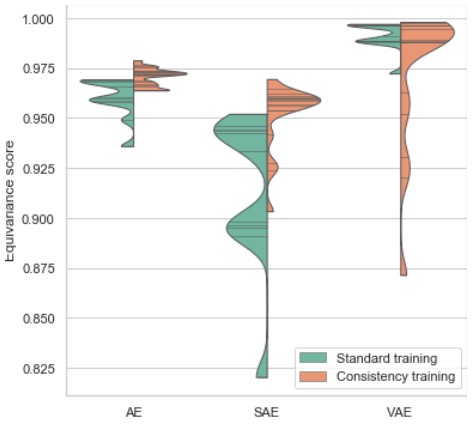

(a) Equivariance score comparing performance of models with and without consistency training.

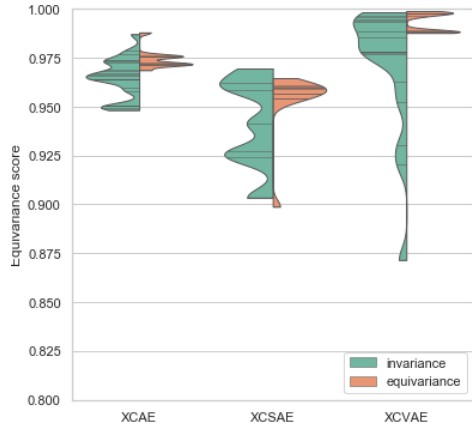

(b) Equivariance score for all the models with consistency training comparing invariance-based and equivariance-based regularisation.

We proceed to present the self-consistency scores for all the experiments. Notice that self-consistency is the only metric that, similarly to Cemgil et al. (2020), directly compares the response to the prior. Since, the AVAE model objective is conceptually analogous to self-consistency, we would expect the AVAE model to score higher than the VAE baseline on this consistency metric. Figure 28 confirms this prediction.

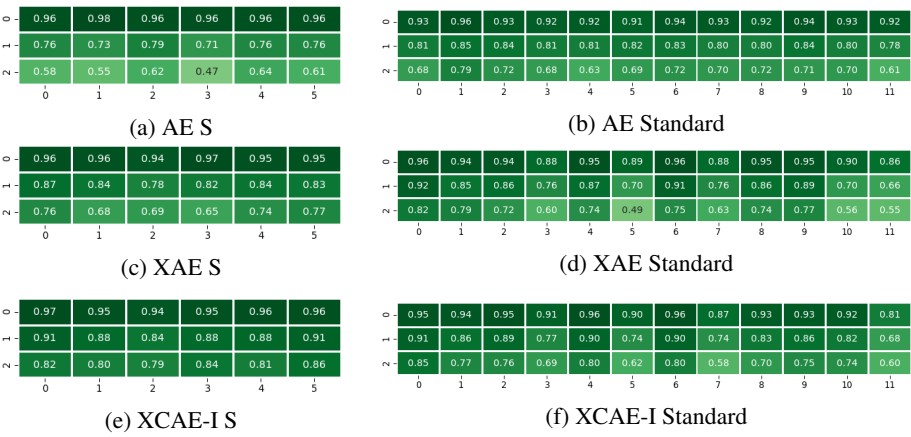

Figure 29: Self-consistency score for each noise variable of the autoencoder family. Results are shown for both the S and Standard model sizes.

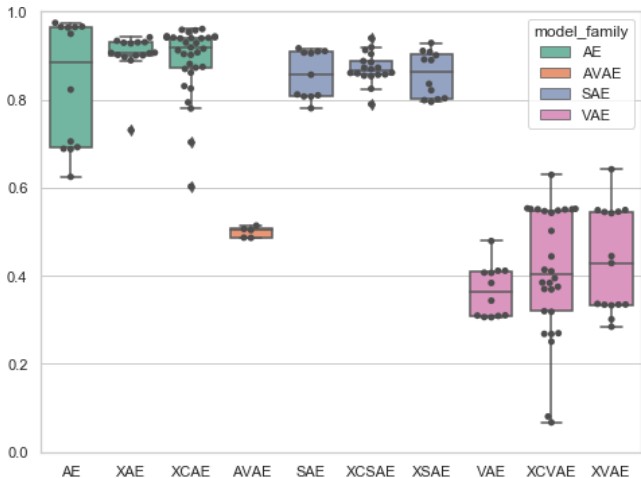

Figure 28: Summary of self-consistency for multiple random seeds and model versions. The score is reduced to a single number by averaging across the dimensions. Note that for the variational models the score is obtained by sampling from the prior distribution, while for the AE and SAE families posterior sampling is used.

### A.4.4 XNETS

**Masks structure.** We can look at the structure learned in the explicit causal latent blocks. The mask values in the Xblock describe a weighted adjacency matrix, which is displayed by the following heatmaps. Interestingly, the structure is not significantly affected by the training scheme adopted (i.e. consistency regularisation) or the model family. Instead its variability is mostly linked to the causal units size and latent size.

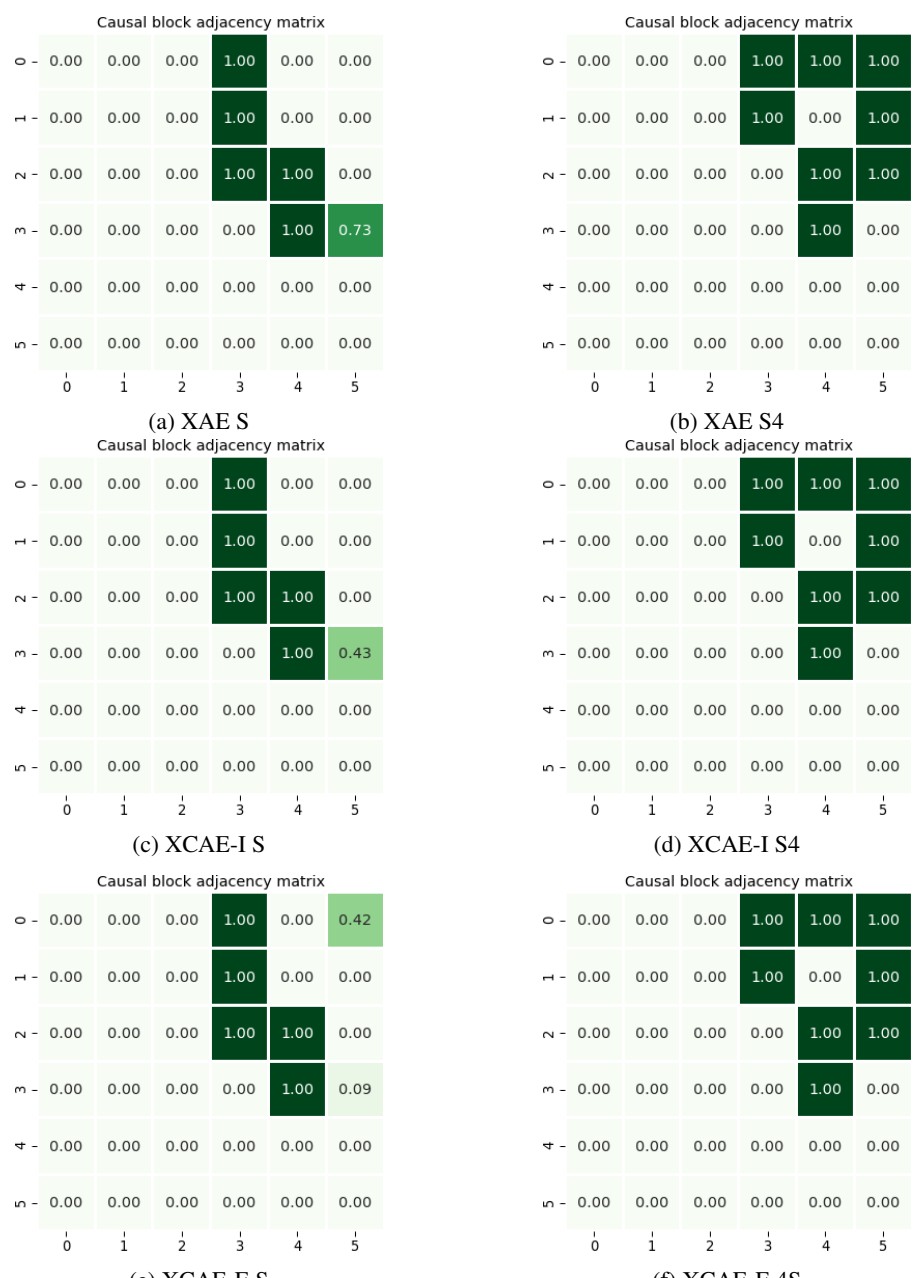

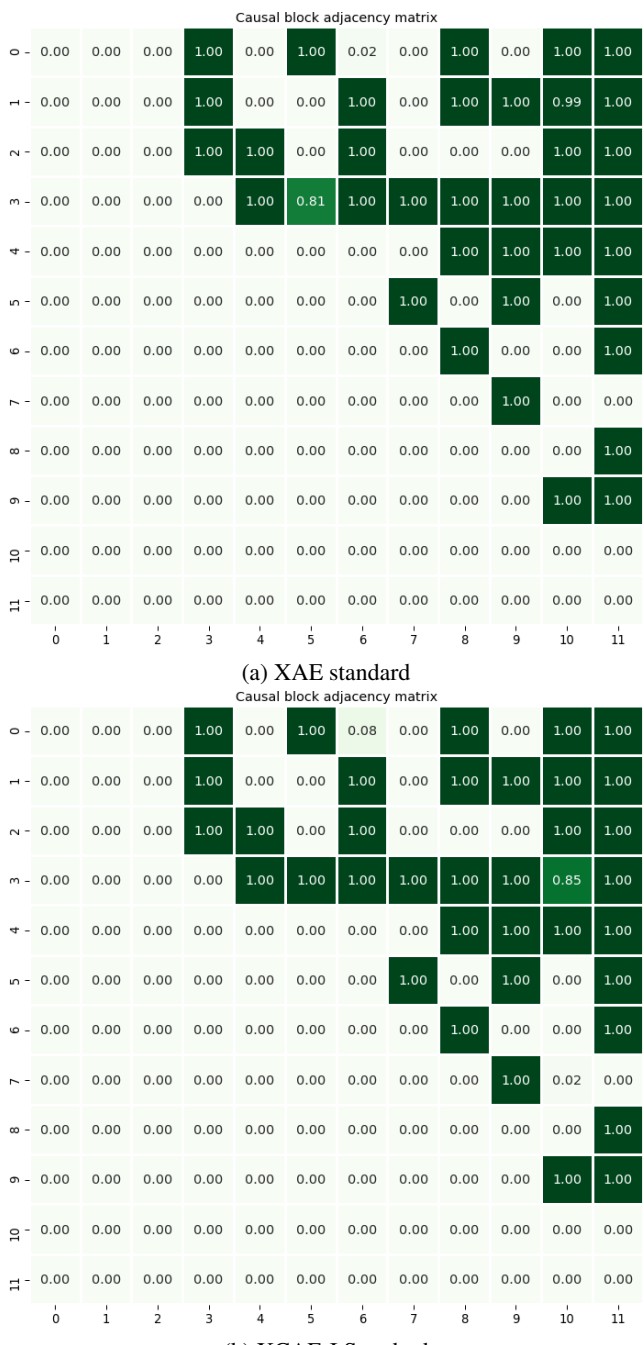

(a) XAE standard

(b) XCAE-I Standard

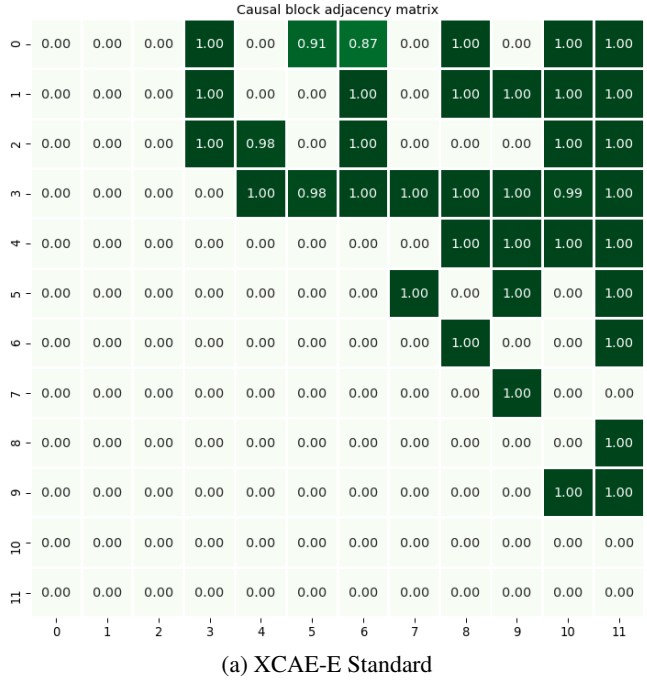

(a) XCAE-E Standard

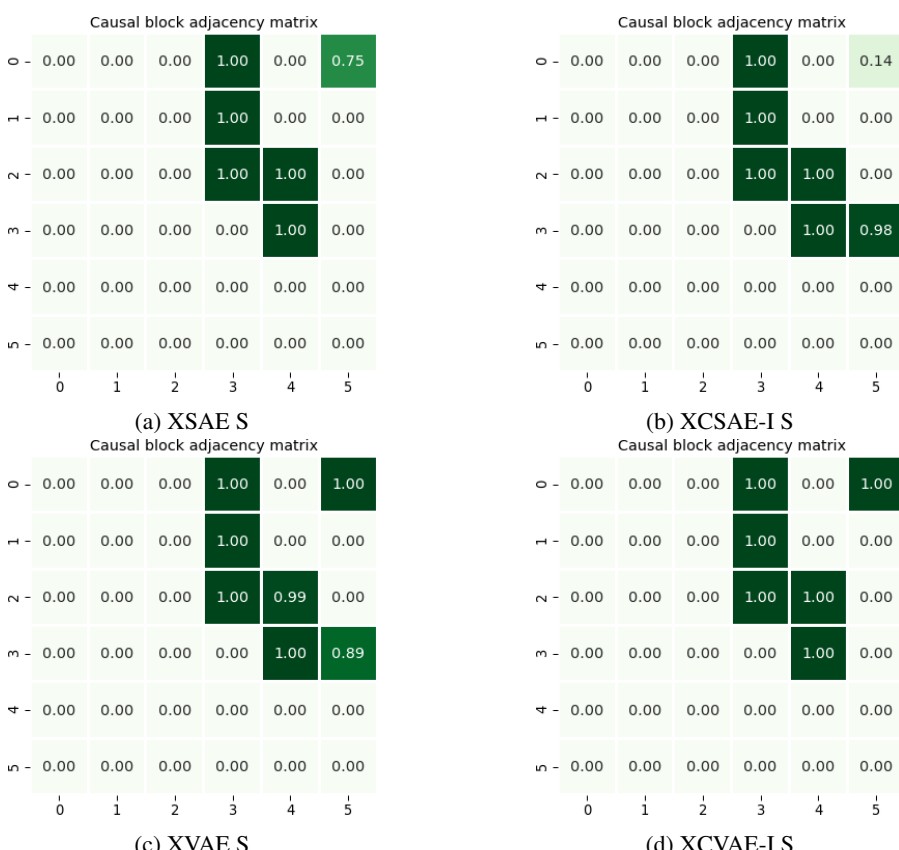

(a) XSAE S

(b) XCSAE-I S

(c) XVAE S

(d) XCVAE-I S

**XBlock visualisation tools.** We introduce a qualitative evaluation tool reliant on interventional consistency that permits to interrogate each representation dimension with specific examples. We

produce hybrids between two or more prior samples by separately intervening on different dimensions. More precisely, given a *base* $N^1 \sim P(N)$ and an *alternative* $N^2 \sim P(N)$, we obtain $n$ hybrids by applying the following interventions one at a time:

$$N_i^1 \leftarrow N_i^2 \ \text{ for } i = 1, ..., n$$

By decoding the result noise vector we obtain a visualisation of the hybridisation: we can thus visually assess the effect of a localised intervention on the representation. Examples are given in Figure 34.

Many dimensions do not show any change in response to the intervention. A lack of reaction could be due to a general insensitivity of the dimension (e.g. the case of a collapsed unit), or to an incidental overlapping of the two samples, or even to an inability of the decoder to generalise outside of support. Moreover for the 'X' and 'XC' model variants we observe that the change produced by the interventions is semantically different on distinct latent dimensions. This suggests that the representation has developed a certain degree of modularity.

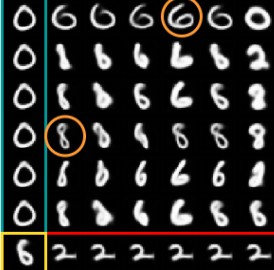 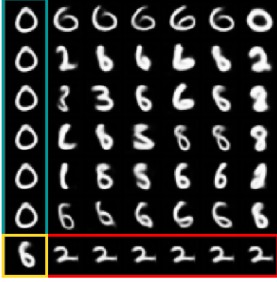

Figure 35: Double intervention matrix on $N$ (left) and $Z$ (right) for a XCAE-I S model on MNIST. The yellow box contains the base sample, while the two alternatives are displayed inside the blue and red boxes. The hybrid samples are displayed in the matrix according to the interventions indices, i.e. element $i, j$ correspond to the hybrid obtained by replacing the $i$-th and $j$-th elements of the base with the corresponding components in the first and second alternative respectively. The items on the diagonal correspond to the result of hybridisation only with the first alternative.

We put consistency training to test by simultaneously applying two interventions on the base sample using two different alternatives. Additionally, we look at the result both when the interventions are applied on the noise and on the causal variables. Applying an intervention on the causes breaks the connection from the parent to the intervened unit, thus altering the statistical relations between causal variables. The results are shown in Figure 35. There is a noticeable difference in the quality of the hybrid samples between the two kinds of interventions: hybrids on the causal variables more often do not appear connected to any of the prior samples, sometimes producing meaningless shapes (e.g. element $(4, 1)$ in the matrix on the right). On the other hand, the noise hybrid samples appear convincing and mostly semantically homogeneous. Interestingly, as highlighted in orange in Figure 35, the two different interventions can provide two different interpretations of the original ambiguous shape.

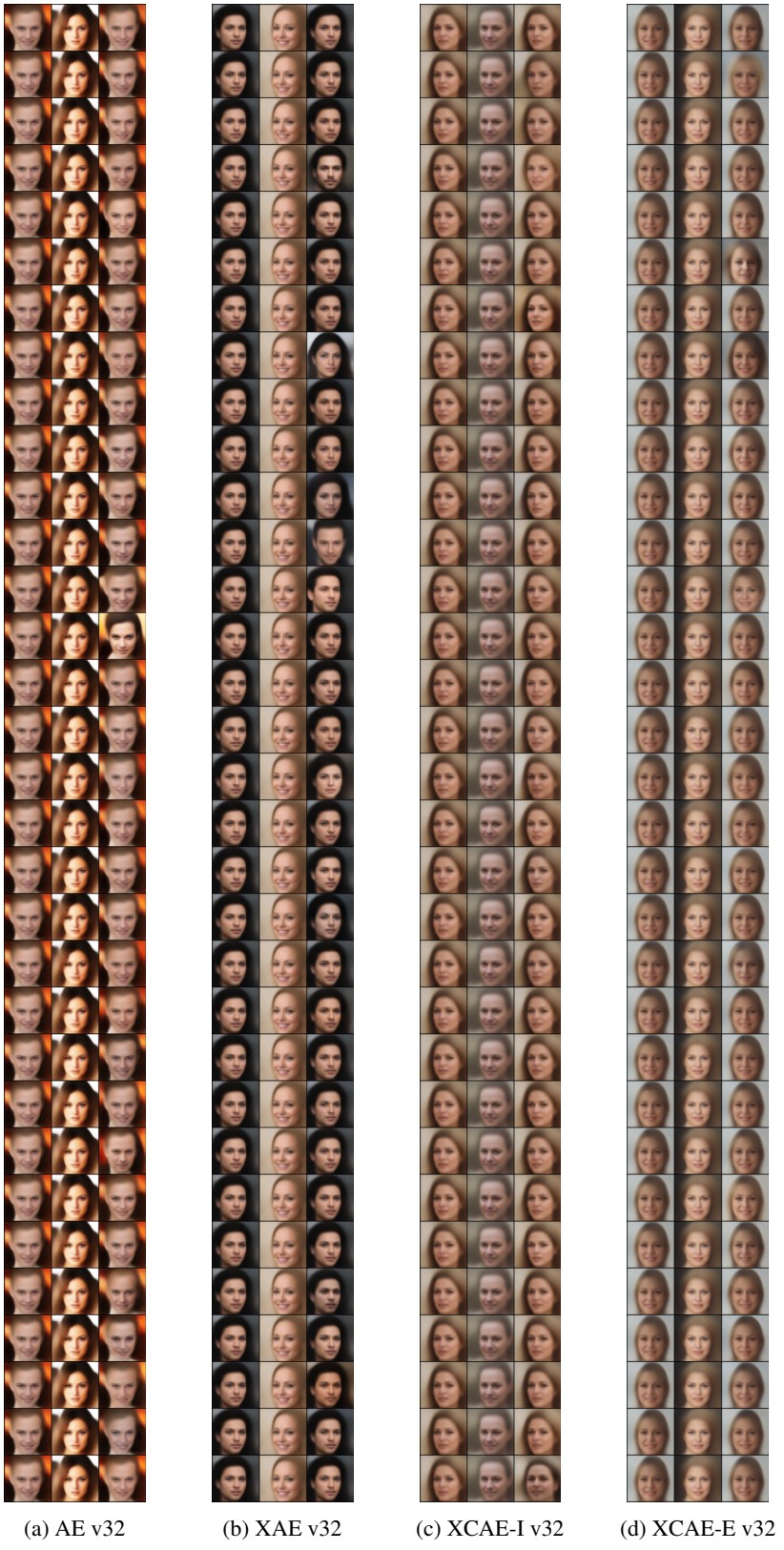

(a) AE v32          (b) XAE v32          (c) XCAE-I v32          (d) XCAE-E v32

Figure 34: Hybridisation on v32 models representation. The left column corresponds to the base sample, the middle column to the presented alternative and the hybridisation result for each dimension is shown on the right.

