# OpenReview forum: "On the interventional consistency of autoencoders"
_ICLR.cc/2022/Conference — ICLR 2022 Submitted_

### Official Review · Reviewer_hhq1 · 2021-10-27

**Correctness:** 3
**Technical Novelty And Significance:** 2
**Empirical Novelty And Significance:** 2
**Recommendation:** 3
**Confidence:** 4

**Main Review:**

- The paper is motivated by the goal of "causal representation learning" and explicitly mentions it does not address the important question of identifiability. The most important problem is that there is no justification as to why the authors' notion of interventional consistency has anything to do with what researchers in the past have meant by a "causal representation" (e.g. disentangled, modular, robust, ...etc.).
- I found the paper hard to read even though the ideas proposed are rather simple. The paper is framed within the larger context of causality, but I am not sure this is the best way to present the contribution, which might explain my confusion.
- The motivation for the "explicit causal latent block" is unclear. Why does it make sense to have the output of the encoder be the noise of the SCM over $Z$?
- Some very important quantity like $\mathcal{L}\_\text{INV}$ and $\mathcal{L}\_\text{EQV}$ (which are the actual regularizer proposed) are defined only in the appendix. Important contributions should be presented in the main text. Also, the explanations for these central quantities lack clarity.
- It is not clear why learning a SCM in the latent space of MNIST makes sense. For instance, I can't imagine the shape of a digit causing its thickness.
- Figure 6 shows three latent traversals for a baseline and two variants of their approach and claim that their approach yield more modular traversal. However, it is not clear what is meant here by modularity and I do not see how the latent traversals of their approach are better.
- The experiments show that indeed the regularizer improves "interventional consistency". But, as I said earlier, it is not clear how this relates to "causal representation learning" or why it is something beneficial regarding disentanglement, robustness, modularity etc.

Minor:

- The notation is sometimes imprecise: for example, in the equation between (2) and (3), it is not clear what is meant by $Z'\_{j \not= i}$. Is it all components of $Z'$ that are not $i$? If so the decomposition is wrong. Is it $Z'\_{j<i}$ for some ordering of the nodes? If so what is the ordering? This point matters because (4) goes on specifying what is meant for the "ICM principle to be preserved by the response map" based on this above equation.
- The notation is sometimes hard to parse: for instance, Section 2.1 has a few double "hat" like this: $\bar{\tilde{Z}}$.
-  Table 1 and 2 have no error bars, making it hard to asses whether the gaps between different methods are significant or not.


**Summary Of The Paper:**

This work proposes the notion of "interventional consistency" as a beneficial property learned representations should have and introduces a regularization term to enforce it in autoencoders. Moreover, the authors introduce an "explicit latent causal block" that allows learning a structural causal model (SCM) over the latent factors. The experimental section argues that both the regularization and the "causal layer" improve interventional consistency.

**Summary Of The Review:**

Given the points raised above, I cannot recommend acceptance. My main concerns are with (i) the lack of motivation for both the "interventional consistency" and the "explicit causal latent block", as well as (ii) the (sometimes significant) lack of clarity. The paper did not make a convincing case that "interventional consistency" is a valuable tool for representation learning.

---

> ### Author Response · Authors · 2021-11-22
> **Response to reviewer hhq1**
>
> Dear reviewer hhq1, thank you for your feedback. We treasure your perspective and we will take all your suggestions into account when revising the presentation for future versions of the paper. We will now address some of the points raised in your review, which we hopefully will be able to clarify for you.
>
> 1. For what concerns the connection with the field of causal representation learning, we frame interventional consistency as a necessary (albeit certainly not sufficient) condition for disentanglement. The link is made explicit in Section 2 (page 2): “Crucially, this property mirrors the definition of disentanglement in the context of the artificial generative process …”. Moreover, throughout the paper we argue that interventional consistency alone renders the representation more modular, which, as you point out, is one of the hallmarks of a ‘causal representation’.
> 2. Moving to the motivation behind the XBlock, the goal is to ‘separate information from structure’ (section 2.2). The bottleneck layer by definition coincides with the output of the encoder: therefore the latent space can be identified as the information source (which in an SCM corresponds to the noise terms). The structural information, which is assumed to be fixed and not dependent on the input, can be captured by the learnable structural mappings applied to the noise variables. The output of the structural assignments are potentially statistically dependent variables, corresponding to the causal variables in an SCM.
> 3. For what concerns the SCM for MNIST, consider that any structure (even a flat structure) is allowed in an SCM. The case of independent latents is a special case of SCM. Therefore instantiating a learnable SCM-like structure in the latent space does not constraint the representation to develop an hierarchical arrangement of its variables. Anyway, whether or not statistical dependencies might be meaningful in the latent space depends on the features included in the representation (which might not always be interpretable). For instance, in the example that you make thickness might be statistically dependent on a third variable representing orientation and both might be caused by a categorical variable depicting the ‘style’ of the drawing.
> 4. Regarding Figure 6, the latent traversals are indeed not generally ‘better’: part of our point (Section 5) is that the traversals and generation quality is kept equal across different model versions. The claim on modularity is mostly informed by the quantitative results discussed later in the section (see Table 2), and, as every qualitative evaluation, it is liable to subjective perception. We acknowledge that the definition of modularity is somewhat blurred and should be elaborated further. What we are referring to here is that the middle and right traversals is that each dimension tends to change the input in different ways, with little overlapping between the affected features. The same does not hold for the left-most samples (see row 4 and row 5).
> 5. Concluding with the notation, we recognise an inconsistency in using $Z’_{i \neq j}$ and $Z’_{i < j}$ interchangeably. $Z’_{i \neq j}$ indeed refers to all the components of $Z’$ that are not $Z_i'$: why would you say that the decomposition is wrong? For $Z’_{i < j}$ the ordering is not predefined. The idea is that interventional consistency should hold for one such factorizations, the one that describes the ‘causal’ process.

---

> > ### Comment · Reviewer_hhq1 · 2021-11-29
> > **Response**
> >
> > I thank the authors for considering my concerns carefully. The lack of clarity, which is still the most important issue, remains and for that reason I will not change my score. Moreover, I find the points 1 & 2 unconvincing. I believe the authors need to clarify these to make their work ready for publication.

---

### Official Review · Reviewer_9iqe · 2021-11-02

**Correctness:** 3
**Technical Novelty And Significance:** 3
**Empirical Novelty And Significance:** 3
**Recommendation:** 6
**Confidence:** 4

**Main Review:**

Strengths
- The idea is well-motivated and seems to be theoretically justified.
- The empirical results are promising and support the main claim of the paper.
- The limitation of the proposed method is also discussed (in the paragraph right before Section 2.1).

Weaknesses
- Presentation of the paper can be improved
  - ICM is not defined when it's firstly used in page 2.
  - There is a broken citation in the 3rd line of the 3rd paragraph of Section 3
  - The first two figures in Section 5 is not numbered.
    - The order of all methods in the two figures is weird/inconsistent. You should keep a consistent order for all baseline + variants.
    - I also suggest better a better way to label them: You can colour each baseline model with a colour and fill the variants with (no filling, dots, dashed lines).

Questions
- For Figure 2, how do I know the traversals are supposed to give low errors? i cannot tell if the traversal is along any disentangled dimension from the images.

**Summary Of The Paper:**

The paper proposes to use interventional consistency to regularise representation learning in VAEs.
The idea is well-motivated by the ICM principle and theoretically justified.
The paper suggests to use interventional consistency for both training and evaluation of the representation learnt by VAEs.
Results show that the proposed idea can give more modular and interpretable representation.

**Summary Of The Review:**

The paper proposes a well-motivated and (seemingly) theoretically justified way to regularise representation learning in VAEs.
The results also support the main claim of the paper.
Therefore I believe this paper is beneficial to the community.

---

> ### Author Response · Authors · 2021-11-22
> **Response to reviewer 9iqe**
>
> Dear reviewer 9iqe, thank you for your feedback and for reporting both the strengths and weaknesses of our work. We appreciate all your suggestions and we will make sure to fix all the issues you raised in a future version of the paper.
>
> For what concerns your question on Figure 2, by definition we expect an interventionally consistent autoencoder to exhibit local responses to local interventions. Thus the response error plots shown in Figure 2 should show low errors on all the dimensions that are not affected by the traversal (which in this specific case is the last dimension). However, for the autoencoder model with standard training (figure on the left) the response to a local intervention perturbs all the features in the representation. We conclude that the model on the left has low interventional consistency. Note that the definition of interventional consistency allows for large errors on the intervened dimension.

---

> > ### Comment · Reviewer_9iqe · 2021-11-29
> > **Thanks for the response**
> >
> > Thanks for the clarification on Figure 2 - it would be nice to include these explanation in the revised maniscript.

---

### Official Review · Reviewer_6BaE · 2021-11-03

**Correctness:** 3
**Technical Novelty And Significance:** 2
**Empirical Novelty And Significance:** 2
**Recommendation:** 5
**Confidence:** 2

**Main Review:**

Frankly, I am not quite convinced why the interventional consistency is desired even after reading the draft. Some assumptions seem too strong to me and I feel the main objective of the paper is not conveyed well.

Some typos and issues listed here:

1) Unexplained notations:

      a) section 2, composition R = E\circle D. here neither E nor D is defined.

      b)  Figure 4 comes before the figure 3

      c) the third paragraph of section 3, related work: there is a citation error with question mark


2) Lack of clarity:

a) The major algorithms 1) and 2) are not described in the main paper.  We have to reach out to the appendix for the algorithm description.

b) The optimization objective L_{INV} and L_{EQV} are not specified in the main paper either.

c) The response map is mainly introduced by citing a related work. However, given its importance to this paper it is better explained and self-contained.

d) There is not much explanation on the latent traversals.


3) Assumptions:

There is an assumption that the statistical dependencies in the prior are preserved by the response map.  This assumption looks to me very strong. Not sure if such an assumption makes sense for real data.


**Summary Of The Paper:**

The authors argue that the interventional consistency is a desired property for the representation of auto-encoders and proposed a new metric on it. The authors further develop a new training procedure for auto-encoder that has extra regularization terms to enforce the interventional consistency.

**Summary Of The Review:**

I feel the draft is not in a status for publication yet. The authors may need to polish it to have a clear story line. Also it needs to be self-contained.


########Update after the authors' rebuttal###########

I would like to thank the authors' explanations on the notations as well as the assumption concerns. Overall I feel this draft has potential to be a good one. As stated in the authors feedback, in its current status more work is needed to make it easy to read. (For example, the algorithm descriptions are still hidden in the appendix.) As such I am keeping my ratings unchanged.

---

> ### Author Response · Authors · 2021-11-22
> **Response to reviewer 6BaE**
>
> Dear reviewer 6BaE, we thank you for your feedback and suggestions. We regret the lack of clarity in our exposition and we are committed to a substantial revision thereof that would allow future readers to appreciate the value of the work.  However, there are a few of the concerns raised in your review that can perhaps be resolved directly in this answer. In order:
>
> - 1.a) $E$ and $D$ have been defined in the Introduction section. Quoting from page 1 “Mathematically autoencoders can be represented as the tuple $(E: \mathbb{R}^d \rightarrow  \mathbb{R}^n, D : \mathbb{R}^n \rightarrow  \mathbb{R}^d)$.“
> - 2.c) The response map is simply defined as the composition $E \circ D$. The citation is only due to our choice to adopt the same terminology. There is not really much to introduce.
> - 2.d) Latent traversals are nowadays a widespread evaluation tool in representation learning, therefore it would appear reasonable to assume familiarity of the reader with the concept. Nonetheless, in the Method section (page 3) we introduce the notion as follows: “[the tool of latent traversals] consists of atomic interventions applied to the different dimensions of the representation.”
> - 3) You are certainly correct in stating that the assumption is generally strong. However, consider the two following points: (i) the optimal solution to the disentanglement problem is supposed to identify the data generative factors and reproduce their statistical dependencies in the latent space, therefore the assumption holds for the optimal solution; (ii) in the case of an Xnet the assumption reduces to joint independence of the noise terms, and in the results’ analysis (Section 5) we make the case that the XBlock increases independence between the noises (see Table 2 and Figure 8).
>
> Having clarified these points, we consider all your suggestions valid and useful. For what concerns the appendix material, would it seem more reasonable to leave out of the main paper the discussion on the approximations and evaluation metrics (i.e. Section 2.1) and include algorithm 1 and 2 and the regularisation terms instead?

---

### Official Review · Reviewer_zHvy · 2021-11-03

**Correctness:** 3
**Technical Novelty And Significance:** 4
**Empirical Novelty And Significance:** 4
**Recommendation:** 3
**Confidence:** 3

**Main Review:**

Strengths
- The proposed concept, the interventional consistency, seems highly novel. It extends the notion of consistency in a meaningful way.

Weaknesses
- The largest weakness of the paper is the lack of clarity. The paper is very difficult to follow, and I believe the difficulty is attributed to the unpolished exposition. In general, Section 2 assumes the readers have a strong background in causal representation learning to an unnecessary extent.
  - The ICM principle is introduced without stating its full name (independent causal mechanisms, I guess?).
  - Since ICM works as a fundamental underlying principle for the work, it deserves a more detailed illustration and justification.
  - It would be helpful for readers if it is stated that PA states "parents".
  - What is the definition of the intervention distribution? What are the examples? What properties of them are assumed?
  - Providing a concrete, real-world example of interventional consistency would be helpful.
  - While the paper says $m$ and $k$ are assumed to be equivalent, Figure 1 depicts the scenario where $m$ and $k$ are different. Additionally, I couldn't come up with a meaningful real-world case where $m\neq k$.
- The other parts of the paper also have clarity issues.
  - Texts in Figure 2, Figure 3, Figure 5, Figure 8 are too small and difficult to read.
  - The figure numbering is missing for Figure 5.
  - There is a broken reference link on page 6.

**Summary Of The Paper:**

The paper proposes a novel principle for representation learning using an autoencoder, named interventional consistency. The tractable objective functions, namely Invariance Score and Equivariance Score, are formulated, and Explicit Causal Latent Block is proposed to model the causal structure of the latent representation.

**Summary Of The Review:**

I vote to reject the paper, due to the lack of clarity. To improve the clarity of the paper, significant rewriting seems to be needed.

---

> ### Author Response · Authors · 2021-11-22
> **Response to reviewer zHvy**
>
> Dear reviewer zHvy, thank you for your feedback. You correctly point out that the conceptual tools from Causality and Causal Representation Learning used in the work lack a proper introduction. We will proceed to reserve more space in the paper to cover the background. For what concerns providing a real-world example, could you perhaps elaborate more on the suggestion? In the presentation we examine the effects of interventional consistency on the latent traversals: this analysis aims precisely at offering a tangible illustration of the application of the principle. Finally, we consider the case of $m \neq k$ for completeness in the formulation: the idea is that $m = k$ is not a necessary condition for interventional consistency. Additionally, the link with disentanglement is more evident if permutation is allowed. However, we agree with you that in the context of autoencoding there is no meaningful real-world case where $m \neq k$, and, to avoid any confusion we will include this observation in the presentation or rather directly consider the case $m = k$ from the start.

---

> > ### Comment · Reviewer_zHvy · 2021-11-30
> > **Thank you for your response**
> >
> > Dear authors,
> >
> > I appreciate your comments.
> >
> > 1. I believe the paper could be an impactful paper if equipped properly with a gentle guide to background knowledge and a clear exposition.
> > 2. I asked about real-world examples because I was wondering how scalable the proposed approach would be. Many challenges in machine learning are closely related to scalability and the paper would become much stronger if the authors can demonstrate the proposed approach is scalable and extensible.
> > 3. I understand the intention of the authors to encompass a more general setting where $m\neq k$.
> >
> > Again, thank you for your submission and response.

---

### Author Response · Authors · 2021-11-20
**Overall response by authors**

Dear reviewers, thank you for your valuable feedback and the time taken to evaluate our work. From your comments we conclude that the paper storyline has to be substantially revised to convey the actual value of the work. Your notes help us identify which parts of the presentation are generating confusion. Our intention is to gather all your suggestions and use them to compile a new version of the paper. To this end, we will provide clarifications on the doubts expressed in the individual reviews wherever additional discussion may be instructive. We are pleased to find that our contribution is considered significant and our approach novel. Since the principal flaw of the paper has been concertedly found to be its lack of clarity, we are confident that changes in the exposition and narrative, informed by your accurate analysis, can render the draft ready for future publication.

---

### Decision · Program_Chairs · 2022-01-20

**Decision:**

Reject

**Comment:**

The paper introduces the notion of interventional consistency of a representation learned using autoencoders, which is claimed to be a desirable property for disentanglement. The reviewers agree that the contributions are novel and relevant, but they also found the paper hard to follow due to a lack of clarity and motivation. Further, they considered the underlying assumptions very strong and possibly hard to find practical instances where they may hold (e.g., the assumption that statistical dependencies in the prior are preserved by the response map). The reviewers also noted that some real-world examples showing the interventional consistency would be helpful.

After all, the paper contains interesting ideas and we would like to encourage the authors to pursue this line of work. Still, the paper in its current form is not ready for publication. We encourage the authors to address the reviewers' comments explicitly in a future version of the manuscript.